

# Qualitative reachability for open interval Markov chains

Jeremy Sproston

Dipartimento di Informatica, University of Turin, Turin, Italy

## ABSTRACT

Interval Markov chains extend classical Markov chains with the possibility to describe transition probabilities using intervals, rather than exact values. While the standard formulation of interval Markov chains features closed intervals, previous work has considered open interval Markov chains, in which the intervals can also be open or half-open. In this article we focus on qualitative reachability problems for open interval Markov chains, which consider whether the optimal (maximum or minimum) probability with which a certain set of states can be reached is equal to 0 or 1. We present polynomial-time algorithms for these problems for both of the standard semantics of interval Markov chains. Our methods do not rely on the closure of open intervals, in contrast to previous approaches for open interval Markov chains, and can address situations in which probability 0 or 1 can be attained not exactly but arbitrarily closely.

## INTRODUCTION

The development of modern computer systems can benefit substantially from a verification phase, in which a formal model of the system is verified exhaustively in order to identify undesirable errors or inefficiencies. In this article we consider the verification of probabilistic systems, in which state-to-state transitions are accompanied by probabilities that specify the relative likelihood with which the transitions occur, using model-checking techniques; see *Baier & Katoen (2008)*, *Forejt et al. (2011)* and *Baier et al. (2018)* for general overviews of this field. One drawback of classical formalisms for probabilistic systems is that they require typically the specification of exact probability values for transitions: in practice, it is likely that such precise information concerning the probability of system behaviour is not available. A solution to this problem is to associate intervals of probabilities with transitions, rather than exact probability values, leading to *interval Markov chains* (IMCs) or *interval Markov decision processes*. IMCs have been studied in *Jonsson & Larsen (1991)* and *Kozine & Utkin (2002)*, and considered in the *qualitative* and *quantitative* model-checking context in *Sen, Viswanathan & Agha (2006)*, *Chatterjee, Sen & Henzinger (2008)* and *Chen, Han & Kwiatkowska (2013)*. Qualitative model checking concerns whether a property is satisfied by the system model with probability (equal to or strictly greater than) 0 or (equal to or strictly less than) 1, whereas quantitative model checking considers whether a property is satisfied with probability (strictly or non-strictly) above or below some

Corresponding author
Jeremy Sproston, sproston@di.unito.it

threshold in the interval $[0, 1]$, and generally involves the computation of the probability of property satisfaction, which is then compared to the aforementioned threshold.

In *Sen, Viswanathan & Agha (2006)*, *Chatterjee, Sen & Henzinger (2008)* and *Chen, Han & Kwiatkowska (2013)*, the intervals associated with transitions are *closed*. This limitation was adressed in *Chakraborty & Katoen (2015)*, which considered the possibility of utilising *open* (and half-open) intervals, in addition to closed intervals. Example of such open IMCs are shown in Figs. 1 and 2. In *Chakraborty & Katoen (2015)*, it was shown that the probability of the satisfaction of a property in an open IMC can be approximated arbitrarily closely by a standard, closed IMC obtained by changing all (half-)open intervals featured in the model to closed intervals with the same endpoints. However, as noted in *Chakraborty & Katoen (2015)*, closing the intervals can involve the loss of information concerning exact solutions. Take, for example, the open IMC in Fig. 1: changing the intervals from $(0, 1)$ to $[0, 1]$ on both of the transitions leaving state $s_0$ means that the minimum probability of reaching the state $s_1$ after starting in state $s_0$ becomes 0, whereas the probability of reaching $s_1$ from $s_0$ is strictly greater than 0 for all ways of assigning probabilities to the transitions in the original IMC.

In this article we propose verification methods for qualitative reachability properties of open IMCs. We consider both of the standard semantics for IMCs. The uncertain Markov chain (UMC) semantics associated with an IMC comprises an infinite number of standard Markov chains, where each Markov chain is obtained from the IMC by fixing a probability for each transition, chosen from the transition's interval. In contrast, the interval Markov decision process (IMDP) semantics associates a single Markov decision process (MDP) with the IMC, where the state set is identical to that of the IMC, and where there are available generally an uncountable number of choices from a state, each of which corresponds to an assignment of probabilities belonging to the intervals of the transitions leaving the associated IMC state. The key difference between the two semantics can be summarised by considering the behaviour from a particular state of the IMC: for a Markov chain of the UMC semantics, the *same* probability distribution over outgoing transitions must *always* be used from the state, whereas in the IMDP semantics the outgoing probability distribution *may change for each visit to the state*. We show that we can obtain exact (not approximate) solutions to qualitative reachability problems for both semantics in polynomial time in the size of the open IMC.

For the UMC semantics, and for three of the four classes of the qualitative reachability problem in the IMDP semantics, the algorithms presented are inspired by methods for qualitative reachability problems of finite MDPs. Certain cases can be dealt with by straightforward reachability analysis on the underlying graph of the IMC. Other cases require the construction of a finite MDP that represents sufficient information regarding the qualitative properties of the IMC. Recall that the classical definition of MDPs (see, for example, *Puterman (1994)* and *Baier & Katoen (2008)*) specifies that an MDP comprises a set of states and a transition relation that associates a number of distributions with each state. A transition from state to state consists of two phases: first a nondeterministic choice between the distributions associated with the source state is made, and second a probabilistic choice to determine the target state is made according to the distribution chosen in the first

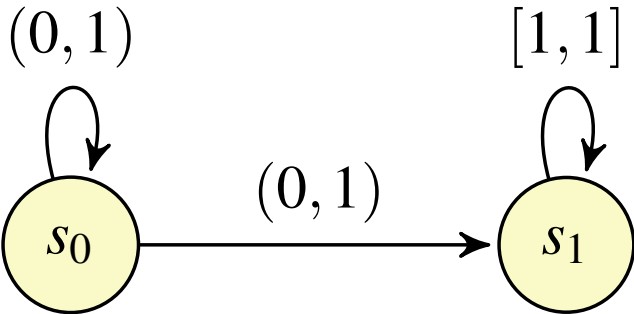

**Figure 1** An open IMC $\mathcal{O}_1$.

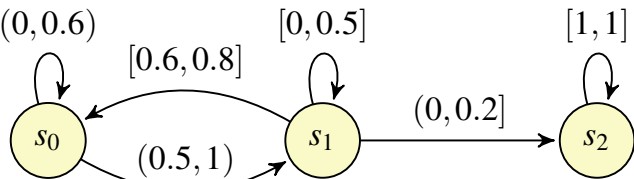

**Figure 2** An open IMC $\mathcal{O}_2$.

phase. The set of states of the finite MDP that we construct equals that of the IMC and, for each state $s$ and each set $X$ of states, a *unique* distribution that assigns positive probability to exactly the states in $X$ is associated with $s$ if and only if there exists at least one probability assignment for target states in the IMC available in $s$ that assigns positive probability to each state in $X$. Intuitively, a distribution associated with $s$ and $X$ in the finite MDP can be regarded as the *representative distribution* of all probability assignments of the IMC that assign positive probability to the transitions from $s$ to states in $X$. For example, in an MDP constructed for the IMC of Fig. 2, there will be two distributions associated with state $s_1$: one distribution assigns positive probability to $s_0$ and $s_2$ (corresponding to the assignment of probability 0.8 to $s_0$, probability 0 to $s_1$, and probability 0.2 to $s_2$), and the other assigns positive probability to $s_0$, $s_1$ and $s_2$ (corresponding to all other possible assignments of probability to the transitions leaving $s_1$, given by the intervals labelling those transitions). Unfortunately, such a finite MDP construction does not yield polynomial-time algorithms in the size of the open IMC, because the presence of transitions having zero as their left endpoint means that the number of representative distributions can be exponential in the number of IMC transitions. In our methods, apart from considering issues concerning the difference between closed and open intervals and the subsequent implications for qualitative reachability problems, we avoid such an exponential blow up. In particular, we show how the predecessor operations used by some qualitative reachability algorithms for MDPs can be applied directly on the IMC.

The remaining, fourth class of reachability problems in the IMDP semantics concerns determining whether the probability of reaching a certain set of states from the current state

[1] Although a finite-memory scheduler can vary the probability of taking the transition from $s_0$ to $s_1$ on the basis of the number of times $s_0$ has been visited, the fact that the memory of the scheduler is finite means that there is some *minimal* probability $\lambda$ of taking the transition from $s_0$ to $s_1$, and hence the overall probability of reaching $s_1$ is at least $\lim_{k \to \infty} 1 - (1 - \lambda)^k = 1$.

is equal to 1 for all schedulers, where a scheduler resolves nondeterminism by choosing an outgoing probability distribution from a state on the basis of the choices made so far. For this class of problems, retaining the memory of previous choices can be important for showing that the problem is *not* satisfied, *i.e.*, that there exists a scheduler such that the reachability probability is strictly less than 1. As an example, we can take the open IMC in Fig. 1. Consider the memoryful scheduler that assigns probability $\frac{1}{2^i}$ to the $i$th attempt to take a transition from $s_0$ to $s_1$, meaning that the overall probability of reaching $s_1$ when starting in $s_0$ under this scheduler is $\frac{1}{2} + \frac{1}{2}(\frac{1}{4} + \frac{3}{4}(\frac{1}{8} + \cdots)) < 1$. Instead a memoryless scheduler will reach $s_1$ with probability 1: for any $\lambda \in (0, 1)$ representing the (constant) probability of taking the transition from $s_0$ to $s_1$, the overall probability of reaching $s_1$ is $\lim_{k \to \infty} 1 - (1 - \lambda)^k = 1$. Similar reasoning can be used to conclude that any finite-memory scheduler also reaches $s_1$ with probability 1.[1] Hence our results for this class of reachability problems take the inadequacy of memoryless and finite-memory schedulers into account; indeed, while the algorithms presented for all other classes of problems (and all problems for the UMC semantics) proceed in a manner similar to that introduced in the literature for finite MDPs, for this class we present an *ad hoc* algorithm, based on an adaptation of the classical notion of end components (*de Alfaro, 1997*).

Our results also allow us to show that, for both of the considered semantics, the same set of IMC states satisfy analogous reachability problems (*i.e*, problems with the same quantification over Markov chains in the UMC semantics and over schedulers in the IMDP semantics, and the same kind of comparison with 0 or 1), except in the case of the reachability problem concerning whether all Markov chains in the UMC semantics or all schedulers in the IMDP semantics reach a certain set of states with probability 1.

After introducing open IMCs in Section 'Open Interval Markov Chains', the algorithms for the UMC semantics and the IMDP semantics are presented in Section 'Qualitative Reachability: UMC semantics' and Section 'Qualitative Reachability: IMDP semantics', respectively.

*Related work.* Model checking of qualitative properties of Markov chains (see, for example, *Vardi (1985)* and *Courcoubetis & Yannakakis (1995)*) relies on the fact that transition probability values are fixed throughout the behaviour of the system, and does not require that exact probability values are taken into account during analysis. The majority of work on model checking for IMCs considers the more general quantitative problems: *Sen, Viswanathan & Agha (2006)* and *Chatterjee, Sen & Henzinger (2008)* present algorithms utilising a finite MDP construction based on encoding extremal probabilities allowed from a state (known as the state's basic feasible solutions) within distributions available from that state. Such a construction results in an exponential blow up, which is also incurred in *Chatterjee, Sen & Henzinger (2008)* for qualitative properties (when transitions can have 0 as their left endpoint). *Chen, Han & Kwiatkowska (2013)* and *Puggelli et al. (2013)* improve on these results to present polynomial-time algorithms for reachability problems based on linear or convex programming. *Haddad & Monmege (2018)* includes polynomial-time methods for computing (maximal) end components, and for computing a single step of value iteration, for interval MDPs. We note that IMCs are a special case of constraint Markov chains (*Caillaud et al., 2011*), and that the UMC semantics of IMCs corresponds

to a special case of parametric Markov chains (*Daws, 2004*; *Lanotte, Maggiolo-Schettini & Troina, 2007*; *Junges et al., 2021*). In particular, the polynomial-time algorithms for positive probability qualitative reachability problems for parametric Markov decision processes in *Junges et al. (2021)* have similarities with our algorithms for the analogous problems for the UMC semantics of IMCs: both are based on the principle that assigning to transitions probability 0, as opposed to positive probability, can never be beneficial from the point of view of satisfying a positive probability reachability problem, an observation which allows us to avoid the aforementioned exponential blow up. *Bart et al. (2018)* present an IMDP semantics that uses finite-memory schedulers, which (as indicated above) is a key difference with our work. As far as we are aware, only (*Chakraborty & Katoen, 2015*) considers open IMCs. This article is a revised and extended version of the conference paper (*Sproston, 2018a*) and the preprint (*Sproston, 2018b*), and includes a detailed treatment of the connections between assignments of probabilities to intervals and syntactic properties of open IMCs, in addition to full proofs of all results.

## PRELIMINARIES

A *(probability) distribution* over a countable set $Q$ is a function $\mu : Q \to [0, 1]$ such that $\sum_{q \in Q} \mu(q) = 1$. Let $\mathsf{Dist}(Q)$ be the set of distributions over $Q$. We use $\mathsf{support}(\mu) = \{q \in Q | \mu(q) > 0\}$ to denote the *support set* of $\mu$, *i.e.*, the set of elements assigned positive probability by $\mu$, and use $\{q \mapsto 1\}$ to denote the distribution that assigns probability 1 to the single element $q$. Given a binary function $f : Q \times Q \to [0, 1]$ and element $q \in Q$, we denote by $f(q, \cdot) : Q \to [0, 1]$ the unary function such that $f(q, \cdot)(q') = f(q, q')$ for each $q' \in Q$.

Let $\mathbb{Q}$ be the set of rational numbers. We let $\mathscr{I}$ denote the set of (open, half-open or closed) intervals that are subsets of $[0, 1]$ and that have rational-numbered endpoints (*i.e.*, endpoints in $\mathbb{Q} \cap [0, 1]$). Given an interval $I \in \mathscr{I}$, we let $\mathsf{left}(I)$ (respectively, $\mathsf{right}(I)$) be the left (respectively, right) endpoint of $I$. The set of closed (respectively, left-open, right-closed; left-closed, right-open; open) intervals in $\mathscr{I}$ is denoted by $\mathscr{I}^{[\cdot, \cdot]}$ (respectively, $\mathscr{I}^{(\cdot, \cdot]}$; $\mathscr{I}^{[\cdot, \cdot)}$; $\mathscr{I}^{(\cdot, \cdot)}$). Note that $\mathscr{I}^{[\cdot, \cdot]}$, $\mathscr{I}^{(\cdot, \cdot]}$, $\mathscr{I}^{[\cdot, \cdot)}$ and $\mathscr{I}^{(\cdot, \cdot)}$ form a partition of $\mathscr{I}$. Let $\langle \cdot, \cdot ]$ be the set of right-closed intervals in $\mathscr{I}$, and let $\mathscr{I}^{\langle 0, \cdot \rangle}$ be the set of intervals in $\mathscr{I}$ with left endpoint equal to 0. We partition $\mathscr{I}^{\langle 0, \cdot \rangle}$ into the set $\mathscr{I}^{[0, \cdot \rangle}$ of left-closed intervals with the left endpoint equal to 0, and the set $\mathscr{I}^{(0, \cdot \rangle}$ of left-open intervals with the left endpoint equal to 0. Furthermore, let $\mathscr{I}^{\langle +, \cdot \rangle} = \mathscr{I} \setminus \mathscr{I}^{\langle 0, \cdot \rangle}$ be the set of intervals such that the left endpoint is positive.

A *discrete-time Markov chain* (DTMC) $\mathscr{D}$ is a pair $(S, \mathbf{P})$ where $S$ is a set of *states*, and $\mathbf{P} : S \times S \to [0, 1]$ is a *transition probability matrix*, such that, for each state $s \in S$, we have $\sum_{s' \in S} \mathbf{P}(s, s') = 1$. Note that $\mathbf{P}(s, \cdot)$ is a distribution, for each state $s \in S$. Intuitively, given states $s, s' \in S$, a transition from $s$ to $s'$ is made with probability $\mathbf{P}(s, s')$. A *path* of DTMC $\mathscr{D}$ is a sequence $s_0 s_1 \cdots$ such that $\mathbf{P}(s_i, s_{i+1}) > 0$ for all $i \geq 0$. Given a path $\rho = s_0 s_1 \cdots$ and $i \geq 0$, we let $\mathsf{state}[\rho](i) = s_i$ be the $(i+1)$-th state along $\rho$. The set of paths of $\mathscr{D}$ starting in state $s \in S$ is denoted by $Paths_\infty^{\mathscr{D}}(s)$. In the standard manner (see, for example, *Baier & Katoen (2008)* and *Forejt et al. (2011)*), given a state $s \in S$, we can define a probability measure $Pr_s^{\mathscr{D}}$ over $Paths_\infty^{\mathscr{D}}(s)$.

A *Markov decision process* (MDP) $\mathcal{M}$ is a pair $(S, \Delta)$ where $S$ is a finite set of *states* and $\Delta : S \to 2^{\mathsf{Dist}(S)}$ is a *transition function* such that $\Delta(s) \neq \emptyset$ for all $s \in S$. An MDP is *finite* if $\Delta(s)$ is finite for all $s \in S$. Intuitively, a transition from state $s$ to some target state is chosen according to two phases: first, a nondeterministic choice is made between distributions in $\Delta(s)$; second, a probabilistic choice between target states is made according to the distribution chosen in the first phase.

A(n infinite) path of an MDP $\mathcal{M}$ is a sequence $s_0 \mu_0 s_1 \mu_1 \cdots$ such that $\mu_i \in \Delta(s_i)$ and $\mu_i(s_{i+1}) > 0$ for all $i \geq 0$. Given a path $\rho = s_0 \mu_0 s_1 \mu_1 \cdots$ and $i \geq 0$, we let $\mathsf{state}[\rho](i) = s_i$ be the $(i+1)$-th state along $\rho$, and let $\mathsf{dist}[\rho](i) = \mu_i$ be the $(i+1)$-th distribution along $\rho$. A finite path is a sequence $r = s_0 \mu_0 s_1 \mu_1 \cdots \mu_{n-1} s_n$ such that $\mu_i \in \Delta(s_i)$ and $\mu_i(s_{i+1}) > 0$ for each $0 \leq i < n$. Let $last(r) = s_n$ denote the final state of $r$. For distribution $\mu \in \Delta(s_n)$ and state $s' \in S$ such that $\mu(s') > 0$, we write $r \mu s'$ to denote the finite path $s_0 \mu_0 s_1 \mu_1 \cdots \mu_{n-1} s_n \mu s'$. We say that $r$ is a *prefix* of the infinite path $\rho$ if $\mathsf{state}[\rho](i) = s_i$ for each $0 \leq i \leq n$, and $\mathsf{dist}[\rho](i) = \mu_i$ for each $0 \leq i < n$. Let $Paths_*^{\mathcal{M}}$ be the set of finite paths of the MDP $\mathcal{M}$. Let $Paths_\infty^{\mathcal{M}}(s)$ and $Paths_*^{\mathcal{M}}(s)$ be the sets of infinite paths and finite paths, respectively, of $\mathcal{M}$ starting in state $s \in S$.

A *scheduler* is a mapping $\sigma : Paths_*^{\mathcal{M}} \to \mathsf{Dist}(\bigcup_{s \in S} \Delta(s))$ such that $\sigma(r) \in \mathsf{Dist}(\Delta(last(r)))$ for each $r \in Paths_*^{\mathcal{M}}$. Let $\Sigma^{\mathcal{M}}$ be the set of schedulers of the MDP $\mathcal{M}$. Given a state $s \in S$ and a scheduler $\sigma$, we can define a countably infinite-state DTMC $\mathscr{D}_s^\sigma$ that corresponds to the behaviour of the scheduler $\sigma$ from state $s$. Formally, the DTMC $\mathscr{D}_s^\sigma = (Paths_*^{\mathcal{M}}(s), \mathbf{P}_s^\sigma)$ has $Paths_*^{\mathcal{M}}(s)$ as its state set, and where its transition probability matrix $\mathbf{P}_s^\sigma$ is defined in the following way: for finite path $r \in Paths_*^{\mathcal{M}}(s)$, distribution $\mu \in \Delta(last(r))$ and state $s' \in S$, we have that $\mathbf{P}_s^\sigma(r, r\mu s') = \sigma(last(r))(\mu) \cdot \mu(s')$. The DTMC $\mathscr{D}_s^\sigma$ can be used to define a probability measure $\mathrm{Pr}_s^\sigma$ over $Paths_\infty^{\mathcal{M}}(s)$ in the standard manner (see *Baier & Katoen (2008)* and *Forejt et al. (2011)*).

A scheduler $\sigma \in \Sigma^{\mathcal{M}}$ is *memoryless* if, for finite paths $r, r' \in Paths_*^{\mathcal{M}}$ such that $last(r) = last(r')$, we have $\sigma(r) = \sigma(r')$. In the sequel, a memoryless scheduler will generally be written as the mapping $\sigma : S \to \mathsf{Dist}(\bigcup_{s \in S} \Delta(s))$. Let $\Sigma_{\mathrm{m}}^{\mathcal{M}}$ be the set of memoryless schedulers of $\mathcal{M}$. Note that, for a memoryless scheduler $\sigma \in \Sigma_{\mathrm{m}}^{\mathcal{M}}$, we can construct a finite DTMC $\tilde{\mathscr{D}}^\sigma = (S, \tilde{\mathbf{P}})$ with $\tilde{\mathbf{P}}(s, s') = \sum_{\mu \in \Delta(s)} \sigma(s)(\mu) \cdot \mu(s')$: we call this DTMC the *folded DTMC of* $\sigma$. The probability measures $\mathrm{Pr}_s^\sigma$ and $\mathrm{Pr}_s^{\tilde{\mathscr{D}}^\sigma}$ assign the same probabilities to measurable sets of paths. This can be seen by considering the following. Recall that the scheduler $\sigma$ and state $s$ induce a DTMC $\mathscr{D}_s^\sigma = (Paths_*^{\mathcal{M}}(s), \mathbf{P}_s^\sigma)$. Now consider the smallest relation $\mathscr{R} \subseteq Paths_*^{\mathcal{M}}(s) \times S$ such that $(r, s') \in \mathscr{R}$ if $last(r) = s'$. Note that, for a given $r \in Paths_*^{\mathcal{M}}(s)$ and $s' \in S$, we have $\sum_{\mu \in \Delta(last(r))} \mathbf{P}_s^\sigma(r, r\mu s') = \sum_{\mu \in \Delta(last(r))} \sigma(last(r))(\mu) \cdot \mu(s') = \tilde{\mathbf{P}}(last(r), s')$. This allows us to conclude that $\mathscr{R}$ induces a *probabilistic bisimulation* (*Larsen & Skou, 1991*) (more formally, the equivalence relation induced by $\mathscr{R}$ on the "combined DTMC" obtained by the union of the state spaces of $\mathscr{D}_s^\sigma$ and $\tilde{\mathscr{D}}^\sigma$ and the standard transition function that corresponds directly from $\mathbf{P}_s^\sigma$ and $\tilde{\mathbf{P}}$). Given that probabilistic bisimilar states of DTMCs induce probability measures that assign the same probability to measurable sets of paths (see, for example, Chapter 10 of *Baier & Katoen, 2008*), we conclude that $\mathrm{Pr}_s^\sigma$ and $\mathrm{Pr}_s^{\tilde{\mathscr{D}}^\sigma}$ assign the same probabilities to measurable sets of paths.

**Remark 1** Note that the classical definition of MDPs (*Puterman, 1994*; *Baier & Katoen, 2008*) features *actions* and a transition function mapping states and actions to distributions over states. As above, a transition consists of two phases: the first phase regards the nondeterministic choice of an action, and the second phase regards the probabilistic choice of the unique distribution associated with the current state and the action chosen in the first phase. In our setting, in which we use MDPs to define the IMDP semantics of IMCs, actions have no explicit significance. For this reason, and in accordance with other papers on IMCs (*Sen, Viswanathan & Agha, 2006*; *Chatterjee, Sen & Henzinger, 2008*; *Chen, Han & Kwiatkowska, 2013*), we choose to omit actions and present the transition function as a mapping from states to sets of distributions. □

# OPEN INTERVAL MARKOV CHAINS

In this section, we recall of the definition of (open) interval Markov chains, and introduce notions such as assignments of probability to transitions, well-formed interval Markov chains and edges of interval Markov chains, together with the relationships between these concepts that will be necessary for subsequent technical results. Portions of this text were previously published as part of a preprint (*Sproston, 2018b*).

**Definition 1 (IMCs):** An (open) *interval Markov chain* (IMC) $\mathcal{O}$ is a pair $(S, \delta)$, where $S$ is a finite set of *states*, and $\delta : S \times S \to \mathscr{I}$ is an *interval-based transition function*. □

Intuitively, the probability of making a transition from state $s$ to state $s'$ of an IMC $(S, \delta)$ is a value from the interval $\delta(s, s')$. Given a state $s \in S$, a distribution $\mathbf{a} \in \mathsf{Dist}(S)$ is an *assignment for* $s$ if $\mathbf{a}(s') \in \delta(s, s')$ for each state $s' \in S$.

## Well-formed IMCs

In the sequel, we will consider only IMCs for which there exists at least one assignment for each state. This restriction can be captured by the following syntactic conditions on the transition function. Let $s \in S$ be a state. The *well-formedness conditions for* $s$ are defined as follows:

1. $\sum_{s' \in S} \mathsf{left}(\delta(s, s')) \leq 1$,
2. $\sum_{s' \in S} \mathsf{left}(\delta(s, s')) = 1$ implies that $\delta(s, s')$ is left-closed for all $s' \in S$,
3. $\sum_{s' \in S} \mathsf{right}(\delta(s, s')) \geq 1$, and
4. $\sum_{s' \in S} \mathsf{right}(\delta(s, s')) = 1$ implies that $\delta(s, s')$ is right-closed for all $s' \in S$.

The following proposition establishes the correspondence between the well-formedness conditions and the existence of an assignment, and generalises to open IMCs similar assertions for IMCs with closed intervals only (for example, in Section 4 of *Haddad & Monmege (2018)*).

**Proposition 1** *Let $(S, \delta)$ be an IMC and $s \in S$ be a state. Then the well-formedness conditions for s are satisfied if and only if there exists an assignment for s.* □

**Proof** ($\Rightarrow$) Assume that the well-formedness conditions for $s$ are satisfied. We proceed by defining a function $f : S \to [0, 1]$ such that:

(a) $\mathsf{left}(\delta(s, s')) + f(s') \in \delta(s, s')$ for all $s' \in S$ and
(b) $\sum_{s' \in S} \mathsf{left}(\delta(s, s')) + f(s') = 1$.

Intuitively, the value $f(s')$ is an "offset" from $\mathsf{left}(\delta(s,s'))$ towards $\mathsf{right}(\delta(s,s'))$. Then we let $\mathbf{a} : S \to [0,1]$ be the function defined by setting $\mathbf{a}(s') = \mathsf{left}(\delta(s,s')) + f(s')$ for all $s' \in S$. The function $\mathbf{a}$ is an assignment: condition (a) in the definition of $f$ establishes that $\mathbf{a}(s') \in \delta(s,s')$ for each state $s' \in S$, and condition (b) establishes that $\mathbf{a} \in \mathsf{Dist}(S)$.

In order to define an appropriate $f$, we consider the following cases.

**Case** $\sum_{s' \in S} \mathsf{left}(\delta(s,s')) = \sum_{s' \in S} \mathsf{right}(\delta(s,s')) = 1$. Given that $\mathsf{left}(\delta(s,s')) \geq 0$, $\mathsf{right}(\delta(s,s')) \geq 0$ and $\mathsf{left}(\delta(s,s')) \leq \mathsf{right}(\delta(s,s'))$ for all $s' \in S$, we have that $\mathsf{left}(\delta(s,s')) = \mathsf{right}(\delta(s,s'))$ (*i.e.*, $\delta(s,s')$ is the degenerate interval $[\mathsf{left}(\delta(s,s')), \mathsf{right}(\delta(s,s'))]$) and therefore $\delta(s,s')$ is left- and right-closed for all $s' \in S$. Let $f(s') = 0$ for all $s' \in S$. Hence $\mathsf{left}(\delta(s,s')) + f(s') = \mathsf{left}(\delta(s,s'))$ for all $s' \in S$. Given that $\delta(s,s')$ being left-closed implies that $\mathsf{left}(\delta(s,s')) \in \delta(s,s')$, and that $\sum_{s' \in S} \mathsf{left}(\delta(s,s')) = 1$ for this case, conditions (a) and (b) in the definition of $f$ aref satisfied.

**Case** $\sum_{s' \in S} \mathsf{left}(\delta(s,s')) = 1$ and $\sum_{s' \in S} \mathsf{right}(\delta(s,s')) > 1$. We let $f(s') = 0$ for all $s' \in S$. Given that the antecedent of the well-formedness condition (2) is satisfied for this case, we have that $\delta(s,s')$ is left-closed for all $s' \in S$. Following the reasoning presented for the previous case, conditions (a) and (b) in the definition of $f$ are satisfied.

**Case** $\sum_{s' \in S} \mathsf{left}(\delta(s,s')) < 1$ and $\sum_{s' \in S} \mathsf{right}(\delta(s,s')) = 1$. We let $f(s') = \mathsf{right}(\delta(s,s')) - \mathsf{left}(\delta(s,s'))$ for all $s' \in S$. Hence $\mathsf{left}(\delta(s,s')) + f(s') = \mathsf{right}(\delta(s,s'))$ for all $s' \in S$. We then proceed similarly to the previous cases: well-formedness condition (4) implies that, for all $s' \in S$, the interval $\delta(s,s')$ is right-closed, and therefore $\mathsf{right}(\delta(s,s')) \in \delta(s,s')$, in turn establishing condition (a) in the definition of $f$. Given that $\sum_{s' \in S} \mathsf{right}(\delta(s,s')) = 1$ in this case, condition (b) in the definition of $f$ is satisfied trivially.

**Case** $\sum_{s' \in S} \mathsf{left}(\delta(s,s')) < 1 < \sum_{s' \in S} \mathsf{right}(\delta(s,s'))$ Given $s' \in S$, let $\mathsf{width}(\delta(s,s')) = \mathsf{right}(\delta(s,s')) - \mathsf{left}(\delta(s,s'))$. Then, for each $s' \in S$, we let:

$$f(s') = (1 - \sum_{s'' \in S} \mathsf{left}(\delta(s,s''))) \cdot \frac{\mathsf{width}(\delta(s,s'))}{\sum_{s'' \in S} \mathsf{width}(\delta(s,s''))}.$$

First we show that condition (a) in the definition of $f$ is satisfied. Our task is to show that $\mathsf{left}(\delta(s,s')) + f(s') \in \delta(s,s')$ for all $s' \in S$. Let $s' \in S$, and consider the following two cases.

**Sub-case** $\mathsf{left}(\delta(s,s')) = \mathsf{right}(\delta(s,s'))$. In this sub-case $\mathsf{width}(\delta(s,s')) = 0$ and hence $f(s') = 0$. Showing that $\mathsf{left}(\delta(s,s')) + f(s') \in \delta(s,s')$ then reduces to showing that $\mathsf{left}(\delta(s,s')) \in \delta(s,s')$. Given that this sub-case specifies that $\delta(s,s')$ is the degenerate interval $[\mathsf{left}(\delta(s,s')), \mathsf{right}(\delta(s,s'))]$, we have established that $\mathsf{left}(\delta(s,s')) + f(s') \in \delta(s,s')$.

**Sub-case** $\mathsf{left}(\delta(s,s')) < \mathsf{right}(\delta(s,s'))$. To show that the choice of $f(s')$ satisfies $\mathsf{left}(\delta(s,s')) + f(s') \in \delta(s,s')$, it suffices to show that $0 < f(s') < \mathsf{width}(\delta(s,s'))$. First, we show that $f(s') > 0$. From the definition of the case we have $\sum_{s'' \in S} \mathsf{left}(\delta(s,s'')) < 1$, and hence $1 - \sum_{s'' \in S} \mathsf{left}(\delta(s,s'')) > 0$. The definition of the sub-case specifies that $\mathsf{width}(\delta(s,s')) > 0$. Combining these two facts with the fact that $\mathsf{width}(\delta(s,s'')) \geq 0$ for all $s'' \in S$ establishes that $f(s') > 0$.

Second, we show that $f(s') < \mathsf{width}(\delta(s,s'))$. From our choice of $f(s')$, we need to show that $\frac{1 - \sum_{s'' \in S} \mathsf{left}(\delta(s,s''))}{\sum_{s'' \in S} \mathsf{width}(\delta(s,s''))} < 1$. Noting that the definition of the case specifies that $1 < \sum_{s'' \in S} \mathsf{right}(\delta(s,s''))$, we obtain the following equivalent statements from that fact

and the definition of $\mathsf{width}(\delta(s,s'))$:

$$1 \quad < \quad \sum_{s'' \in S} \mathsf{right}(\delta(s,s''))$$

$$\Leftrightarrow \qquad\qquad 1 \quad < \quad \sum_{s'' \in S} \mathsf{left}(\delta(s,s'')) + \mathsf{width}(\delta(s,s''))$$

$$\Leftrightarrow \qquad 1 - \sum_{s'' \in S} \mathsf{left}(\delta(s,s'')) \quad < \quad \sum_{s'' \in S} \mathsf{width}(\delta(s,s''))$$

$$\Leftrightarrow \qquad \frac{1 - \sum_{s'' \in S} \mathsf{left}(\delta(s,s''))}{\sum_{s'' \in S} \mathsf{width}(\delta(s,s''))} \quad < \quad 1.$$

Hence we have established that the choice of $f(s')$ satisfies $\mathsf{left}(\delta(s,s')) + f(s') \in \delta(s,s')$. Next we show that condition (b) of the definition of $f$ is satisfied. From the choice of $f$ and straightforward rearranging, we have:

$$\sum_{s' \in S} \mathsf{left}(\delta(s,s')) + f(s')$$

$$= \sum_{s' \in S} \mathsf{left}(\delta(s,s')) + \sum_{s' \in S} f(s')$$

$$= \sum_{s' \in S} \mathsf{left}(\delta(s,s')) + \sum_{s' \in S} (1 - \sum_{s'' \in S} \mathsf{left}(\delta(s,s''))) \cdot \frac{\mathsf{width}(\delta(s,s'))}{\sum_{s'' \in S} \mathsf{width}(\delta(s,s''))}$$

$$= \sum_{s' \in S} \mathsf{left}(\delta(s,s')) + (1 - \sum_{s'' \in S} \mathsf{left}(\delta(s,s''))) \cdot \frac{\sum_{s' \in S} \mathsf{width}(\delta(s,s'))}{\sum_{s'' \in S} \mathsf{width}(\delta(s,s''))}$$

$$= \sum_{s' \in S} \mathsf{left}(\delta(s,s')) + (1 - \sum_{s'' \in S} \mathsf{left}(\delta(s,s'')))$$

$$= 1.$$

Hence we have established the ($\Rightarrow$) direction of the theorem. ($\Leftarrow$) Assume that there exists an assignment $\mathbf{a}$ for state $s$. Recall that, from the definition of assignments, we have $\mathbf{a}(s') \in \delta(s,s')$ for all $s' \in S$. Hence $\mathbf{a}(s') \geq \mathsf{left}(\delta(s,s'))$ and $\mathbf{a}(s') \leq \mathsf{right}(\delta(s,s'))$ for all $s' \in S$. Furthermore, given that an assignment is a distribution, we have $\sum_{s' \in S} \mathbf{a}(s') = 1$. From these facts, we obtain $\sum_{s' \in S} \mathsf{left}(\delta(s,s')) \leq \sum_{s' \in S} \mathbf{a}(s') = 1$ and $\sum_{s' \in S} \mathsf{right}(\delta(s,s')) \geq \sum_{s' \in S} \mathbf{a}(s') = 1$, establishing conditions (1) and (3) of well-formedness for $s$. Now consider the case in which $\sum_{s' \in S} \mathsf{left}(\delta(s,s')) = 1$. Given that $\sum_{s' \in S} \mathbf{a}(s') = 1$ and $\mathbf{a}(s') \geq \mathsf{left}(\delta(s,s'))$ for all $s' \in S$, we must have $\mathbf{a}(s') = \mathsf{left}(\delta(s,s'))$ for all $s' \in S$. From the fact that $\mathbf{a}(s') \in \delta(s,s')$ for all $s' \in S$, it must be the case that $\delta(s,s')$ is left-closed for all $s' \in S$. Hence condition (2) of well-formedness for $s$ is established. Following similar reasoning, we can show that, in the case in which $\sum_{s' \in S} \mathsf{right}(\delta(s,s')) = 1$, we have $\delta(s,s')$ is right-closed for all $s' \in S$, thereby establishing condition (4) of well-formedness. ∎

Henceforth we assume that the IMCs that we consider satisfy the well-formedness conditions for each of their states.

**Peer**J Computer Science

[2]The conference version of this article (*Sproston, 2018a*) considered only the first condition in its definition of edges, and hence its results were restricted to the subclass of IMCs for which there were no state pairs $(s, s')$ such that condition (1) holds and condition (2) does not hold.

## Edges of IMCs

In the following, we refer to *edges* as those state pairs to which the transition function can associate positive probability. There are two situations in which the transition function enforces the association of probability 0 to a state pair $(s, s') \in S \times S$: first, when the right endpoint of $\delta(s, s')$ is equal to 0 (and hence $\delta(s, s')$ is the interval $[0, 0]$); second, when the sum of the left endpoints of transitions from $s$ that do *not* have $s'$ as target state is equal to 1 (thereby preventing the assignment of positive probability to $(s, s')$). Hence we define formally the set of edges $E$ of $\mathcal{O}$ as the smallest set such that $(s, s') \in E$ if (1) $\mathsf{right}(\delta(s, s')) > 0$ and (2) $\sum_{s'' \in S \setminus \{s'\}} \mathsf{left}(\delta(s, s'')) < 1$.[2]

We now formalise the intuition that edges are those state pairs which can be assigned positive probability by relating the notions of edges and assignments.

**Proposition 2** *Let $(S, \delta)$ be an IMC and $s, s' \in S$ be states. Then $(s, s') \in E$ if and only if there exists an assignment $\mathbf{a}$ for $s$ such that $\mathbf{a}(s') > 0$.* $\qquad\square$

**Proof** ($\Rightarrow$) Assume that $(s, s)' \in E$. From the definition of the set $E$ of edges, we have $\mathsf{right}(\delta(s, s)') > 0$ and $\sum_{s'' \in S \setminus \{s'\}} \mathsf{left}(\delta(s, s'')) < 1$. We now show that there exists an assignment $\mathbf{a}$ for $s$ such that $\mathbf{a}(s') > 0$ by adopting the construction of $\mathbf{a}$ from the ($\Rightarrow$) direction of Proposition 1; in particular, we note that $\mathbf{a}(s') = \mathsf{left}(\delta(s, s')) + f(s')$ where $f(s') = (1 - \sum_{s'' \in S} \mathsf{left}(\delta(s, s''))) \cdot \frac{\mathsf{width}(\delta(s, s'))}{\sum_{s'' \in S} \mathsf{width}(\delta(s, s''))}$.

Consider the following three cases:

**Case** $\mathsf{left}(\delta(s, s')) = \mathsf{right}(\delta(s, s'))$. Given that $\mathsf{width}(\delta(s, s')) = 0$ in this case, we have $f(s') = 0$. Hence $\mathbf{a}(s') = \mathsf{left}(\delta(s, s')) + f(s') = \mathsf{left}(\delta(s, s')) = \mathsf{right}(\delta(s, s'))$. From $\mathsf{right}(\delta(s, s')) > 0$, we conclude that $\mathbf{a}(s') > 0$.

**Case** $\mathsf{left}(\delta(s, s')) < \mathsf{right}(\delta(s, s'))$ and $\sum_{s'' \in S} \mathsf{left}(\delta(s, s'')) = 1$. By the definition of edges, we have $\sum_{s'' \in S \setminus \{s'\}} \mathsf{left}(\delta(s, s'')) < 1$. Given that $\sum_{s'' \in S} \mathsf{left}(\delta(s, s'')) = 1$ in this case, we have $\mathsf{left}(\delta(s, s')) > 0$. Given that $f(s') \geq 0$, we conclude that $\mathbf{a}(s') > 0$.

**Case** $\mathsf{left}(\delta(s, s')) < \mathsf{right}(\delta(s, s'))$ and $\sum_{s'' \in S} \mathsf{left}(\delta(s, s'')) < 1$. In this case, $\mathsf{width}(\delta(s, s')) > 0$ and $1 - \sum_{s'' \in S} \mathsf{left}(\delta(s, s'')) > 0$. From the fact that $\mathsf{left}(\delta(s, s'')) \geq 0$ for all $s'' \in S$, we have that $f(s') > 0$, and hence $\mathbf{a}(s') > 0$. ($\Leftarrow$) We prove the contrapositive, *i.e.*, we show that $(s, s') \notin E$ implies that all assignments $\mathbf{a}$ for $s$ are such that $\mathbf{a}(s') = 0$. Assume $(s, s') \notin E$. Hence $(1')$ $\mathsf{right}(\delta(s, s')) = 0$ or $(2')$ $\sum_{s'' \in S \setminus \{s'\}} \mathsf{left}(\delta(s, s'')) = 1$ (note that, by well-formedness, we cannot have $\sum_{s'' \in S \setminus \{s'\}} \mathsf{left}(\delta(s, s'')) > 1$). We consider these two cases in turn:

**Case $(1')$:** $\mathsf{right}(\delta(s, s')) = 0$. For all assignments $\mathbf{a}$ for $s$, given that $\mathbf{a}(s') \in \delta(s, s')$, we have $\mathbf{a}(s') \leq \mathsf{right}(\delta(s, s')) = 0$.

**Case $(2')$:** $\sum_{s'' \in S \setminus \{s'\}} \mathsf{left}(\delta(s, s'')) = 1$. Consider an (arbitrary) assignment $\mathbf{a}$ for $s$. Recall that $\mathbf{a}(s'') \geq \mathsf{left}(\delta(s, s'')) \geq 0$ for all $s'' \in S$. Hence $\sum_{s'' \in S \setminus \{s'\}} \mathsf{left}(\delta(s, s'')) = 1$ means that $\sum_{s'' \in S \setminus \{s'\}} \mathbf{a}(s'') \geq 1$. Then, given that assignments are distributions, and hence that $\sum_{s'' \in S} \mathbf{a}(s'') = 1$, we must have $\mathbf{a}(s') = 0$. Because we chose $\mathbf{a}$ to be an arbitrary assignment for $s$, we have shown that $\mathbf{a}(s') = 0$ for all assignments $\mathbf{a}$ for $s$.

Hence we have established that the existence of an assignment $\mathbf{a}$ for $s$ such that $\mathbf{a}(s') > 0$ implies that $(s, s') \in E$. $\qquad\blacksquare$

We use edges to define the notion of path for IMCs: a path of an IMC $\mathcal{O} = (S, \delta)$ is a sequence $s_0 s_1 \cdots$ such that $(s_i, s_{i+1}) \in E$ for all $i \geq 0$. Given a path $\rho = s_0 s_1 \cdots$ and $i \geq 0$, we

let $\mathsf{state}[\rho](i) = s_i$ be the $(i+1)$-th state along $\rho$. We use $Paths_\infty^{\mathscr{O}}$ to denote the set of paths of $\mathscr{O}$, $Paths_*^{\mathscr{O}}$ to denote the set of finite paths of $\mathscr{O}$, and $Paths_\infty^{\mathscr{O}}(s)$ and $Paths_*^{\mathscr{O}}(s)$ to denote the sets of paths and finite paths starting in state $s \in S$. We refer to $(S, E)$ as the *graph of* $\mathscr{O}$.

We define the *size of an IMC* $\mathscr{O} = (S, \delta)$ as the size of the representation of $\delta$, which is the sum over all states $s, s' \in S$ of the binary representation of the endpoints of $\delta(s, s')$, where rational numbers are encoded as the quotient of integers written in binary.

IMCs are presented typically with regard to two semantics, which we consider in turn. Given an IMC $\mathscr{O} = (S, \delta)$, the *uncertain Markov chain* (UMC) semantics of $\mathscr{O}$, denoted by $[\mathscr{O}]_U$, is the smallest set of DTMCs such that $(S, \mathbf{P}) \in [\mathscr{O}]_U$ if, for each state $s \in S$, the distribution $\mathbf{P}(s, \cdot)$ is an assignment for $s$. The *interval Markov decision process* (IMDP) semantics of $\mathscr{O}$, denoted by $[\mathscr{O}]_I$, is the MDP $(S, \Delta)$ where, for each state $s \in S$, we let $\Delta(s)$ be the set of assignments for $s$.

Let $T \subseteq S$ be a set of states of IMC $\mathscr{O} = (S, \delta)$. We define $\mathsf{Reach}(T) \subseteq Paths_\infty^{\mathscr{O}}$ to be the set of paths of $\mathscr{O}$ that reach at least one state in $T$. Formally, $\mathsf{Reach}(T) = \{\rho \in Paths_\infty^{\mathscr{O}} | \exists i \in \mathbb{N}.\mathsf{state}[\rho](i) \in T\}$. In the following we assume without loss of generality that states in $T$ are absorbing in all the IMCs that we consider, *i.e.*, $\delta(s, s) = [1, 1]$ for all states $s \in T$.

### Valid edge sets

Let $\mathscr{O} = (S, \delta)$ be an IMC with edge set $E$, and let $s \in S$ be a state of $\mathscr{O}$. Edge $(s', s'') \in E$ has *source* $s$ if $s = s'$ and *target* $s$ if $s = s''$. Let $E(s) = \{(s, s') \in E | s' \in S\}$ be the set of edges of $\mathscr{O}$ with source $s$. Given $\star \in \{[\cdot, \cdot], (\cdot, \cdot], [\cdot, \cdot), (\cdot, \cdot), \langle +, \cdot \rangle, [0, \cdot), (0, \cdot), \langle 0, \cdot \rangle, \langle \cdot, \cdot ]\}$, let $E^\star = \{(s, s') \in E | \delta(s, s') \in \mathscr{I}^\star\}$. Given $X \subseteq S$, and given $s$ and $\star$ as defined above, let $E(s, X) = \{(s, s') \in E(s) | s' \in X\}$ be the set of edges with source $s$ and target in $X$, and let $E^\star(s, X) = E(s, X) \cap E^\star$.

In the sequel, we will be interested in identifying the sets of edges with source state $s \in S$ and target states that correspond exactly to the set of states assigned positive probability by assignments for $s$. This will allow us to reason about sets of edges with certain characteristics rather than reasoning directly about assignments. The characteristics of sets of edges with the same source state that we consider are the following: the first condition, called largeness, requires that the sum of the upper bounds of the set's edges' intervals is at least 1; the second condition, called realisability, requires that the edges that are *not* included in the set can be assigned probability 0. The formal definition of these characteristics now follows.

**Definition 2 (Large, realisable and valid edge sets.):** *Let* $B \subseteq E(s)$ *be a set of edges with source state* $s \in S$. *The set* $B$ *is:*

- large *if either (a)* $\sum_{e \in B}\mathsf{right}(\delta(e)) > 1$ *or (b)* $\sum_{e \in B}\mathsf{right}(\delta(e)) = 1$ *and* $B \subseteq E^{\langle \cdot, \cdot ]}$;
- realisable *if* $E(s) \setminus B \subseteq E^{[0, \cdot)}$;
- valid *if it is large and realisable.*  □

The following lemma specifies that a valid set of edges with source state $s$ characterises exactly the support sets of at least one assignment for $s$. In its statement and proof we use the following notation. Given $s \in S$ and $B \subseteq E(s)$, we partition $S$ into the following sets:

- $T_B = \{s' | (s, s') \in B\}$ denotes the target states of edges in $B$,

- $T_{E(s)\setminus B} = \{s' \mid (s,s') \in E(s) \setminus B\}$ denotes the target states of edges with source $s$ that are not in $B$, and
- $S_{\neg E(s)} = S \setminus \{s' \mid (s,s') \in E(s)\}$ denotes states that are not target states of the edges that have source state $s$.

**Lemma 1** *Let $s \in S$ and $B \subseteq E(s)$. Then $B$ is valid if and only if there exists an assignment* $\mathbf{a}$ *for $s$ such that* $\mathsf{support}(\mathbf{a}) = T_B$. $\qquad\square$

**Proof** ($\Rightarrow$) Let $B$ be a valid subset of $E(s)$ of edges with source $s$. We have $\sum_{e \in B}\mathsf{left}(\delta(e)) \leq \sum_{e \in E(s)}\mathsf{left}(\delta(e)) \leq \sum_{e \in E}\mathsf{left}(\delta(e)) \leq 1$ from $B \subseteq E(s) \subseteq E$ and condition (1) of well-formedness. Furthermore, $\sum_{e \in B}\mathsf{right}(\delta(e)) \geq 1$ because $B$ is large.

First we identify conditions that are analogues of well-formedness conditions restricted to the set $B$ of edges. Then, using these conditions, we proceed as in the ($\Rightarrow$) direction of Proposition 1 to define an assignment that assigns positive probability only to target states of edges in $B$.

The well-formedness conditions (1) and (2), together with the fact that $B \subseteq E(s)$, establishes the following conditions ($\tilde{1}$) and ($\tilde{2}$), whereas the largeness of $B$ establishes the following conditions ($\tilde{3}$) and ($\tilde{4}$):

($\tilde{1}$) $\sum_{e \in B}\mathsf{left}(\delta(e)) \leq 1$,

($\tilde{2}$) $\sum_{e \in B}\mathsf{left}(\delta(e)) = 1$ implies that $\delta(e)$ is left-closed for all $e \in B$,

($\tilde{3}$) $\sum_{e \in B}\mathsf{right}(\delta(e)) \geq 1$, and

($\tilde{4}$) $\sum_{e \in B}\mathsf{right}(\delta(e)) = 1$ implies that $\delta(e)$ is right-closed for all $e \in B$.

It will be useful to consider the following strengthening of condition ($\tilde{2}$):

($\tilde{2}'$) $\sum_{e \in B}\mathsf{left}(\delta(e)) = 1$ implies that $\delta(s,s')$ is left-closed for all $s' \in S$.

Condition ($\tilde{2}'$) has the following justification. Assume that $\sum_{e \in B}\mathsf{left}(\delta(e)) = 1$. From this fact, and from the combination of $\sum_{s' \in S}\mathsf{left}(\delta(s,s')) \leq 1$ (that is, condition (1) of well-formedness) and the fact that $B \subseteq \{s\} \times S$, we have that $\sum_{s' \in S}\mathsf{left}(\delta(s,s')) = 1$. Then, by condition (2) of well-formedness, we have that $\delta(s,s')$ is left-closed for all $s' \in S$. We note that an analogous strengthening of condition ($\tilde{4}$) cannot in general be obtained (because it is possible that there exists at least one pair $(s,s') \in (\{s\} \times S) \setminus B$ such that $\mathsf{right}(\delta(s,s')) > 0$ without contradicting well-formedness condition (3)).

Next, we define an assignment in a similar manner to that featured in the ($\Rightarrow$) direction of Proposition 1, taking care to guarantee that the assignment we define assigns positive probability only to target states of edges in $B$. As in the ($\Rightarrow$) direction of Proposition 1, we define a function $f : S \to [0,1]$ such that:

(a) $\mathsf{left}(\delta(s,s')) + f(s') \in \delta(s,s')$ for all $s' \in S$ and

(b) $\sum_{s' \in S}\mathsf{left}(\delta(s,s')) + f(s') = 1$.

We define $\mathbf{a} : S \to [0,1]$ by setting $\mathbf{a}(s') = \mathsf{left}(\delta(s,s')) + f(s')$ for all $s' \in S$. As in the proof of Proposition 1, the function $\mathbf{a}$ is an assignment, because condition (a) in the definition of $f$ establishes that $\mathbf{a}(s') \in \delta(s,s')$ for each state $s' \in S$, and condition (b) establishes that $\mathbf{a} \in \mathsf{Dist}(S)$.

In order to define an appropriate $f$, we consider the following cases.

**Case** $\sum_{e \in B} \mathsf{left}(\delta(e)) = 1$. As in analogous cases of the proof of Proposition 1, we let $f(s') = 0$ for all $s' \in S$. Condition (a) in the definition of $f$ is satisfied for the following reason. Given that $\sum_{e \in B} \mathsf{left}(\delta(e)) = 1$, the antecedent of condition $(\tilde{2}')$ is satisfied, and hence $\delta(s, s')$ is left-closed (*i.e.*, $\mathsf{left}(\delta(s, s')) \in \delta(s, s')$) for all $s' \in S$; then the choice of $f(s') = 0$ for all $s' \in S$ means that $\mathsf{left}(\delta(s, s')) + f(s') \in \delta(s, s')$ is satisfied for all $s' \in S$, hence establishing condition (a). We now establish the satisfaction of condition (b) in the definition of $f$. Given that $f(s') = 0$ for all $s' \in S$, showing condition (b) reduces to showing that $\sum_{s' \in S} \mathsf{left}(\delta(s, s')) = 1$. Given that $\sum_{e \in B} \mathsf{left}(\delta(e)) = 1$, our task reduces in turn to showing that $\mathsf{left}(\delta(s, s')) = 0$ for all $s' \in S \setminus T_B$. Recall that $S \setminus T_B = T_{E(s) \setminus B} \cup S_{\neg E(s)}$ (because $T_B$, $T_{E(s) \setminus B}$ and $S_{\neg E(s)}$ form a partition of $S$). We proceed first by showing that $\mathsf{left}(\delta(s, s')) = 0$ for all $s' \in T_{E(s) \setminus B}$. Given that $E(s) \setminus B \subseteq E^{[0, \cdot\rangle}$ from the realisability of $B$, and that $\mathsf{left}(\delta(e)) = 0$ for all $e \in E^{[0, \cdot\rangle}$ by definition, we have that $\mathsf{left}(\delta(s, s')) = 0$ for all $s' \in T_{E(s) \setminus B}$. Next, we show that $\mathsf{left}(\delta(s, s')) = 0$ for all $s' \in S_{\neg E(s)}$. From the definition of edges, for $s' \in S_{\neg E(s)}$, either $\mathsf{right}(\delta(s, s')) = 0$ or $\sum_{s'' \in S \setminus \{s'\}} \mathsf{left}(\delta(s, s'')) \geq 1$. In the case of $\mathsf{right}(\delta(s, s')) = 0$, from $\mathsf{left}(\delta(s, s')) \leq \mathsf{right}(\delta(s, s'))$ (by definition of $\delta$), we must have $\mathsf{left}(\delta(s, s')) = 0$. In the case of $\sum_{s'' \in S \setminus \{s'\}} \mathsf{left}(\delta(s, s'')) \geq 1$, given that well-formedness condition (1) specifies that $\sum_{s'' \in S} \mathsf{left}(\delta(s, s'')) \leq 1$, we must have $\mathsf{left}(\delta(s, s')) = 0$. Hence we have established condition (b) in the definition of $f$. Next we establish that $\mathsf{support}(\mathbf{a}) = T_B$. First we show that $\mathsf{support}(\mathbf{a}) \subseteq T_B$. From the reasoning in the previous paragraph, we have $\mathsf{left}(\delta(s, s')) = 0$ for all $s' \in S \setminus T_B$. Given that $f(s') = 0$ for all $s' \in S$, we have $\mathbf{a}(s') = \mathsf{left}(\delta(s, s')) + f(s') = 0$ for all $s' \in S \setminus T_B$. Hence $\mathsf{support}(\mathbf{a}) \subseteq T_B$. Second we show that $\mathsf{support}(\mathbf{a}) \supseteq T_B$. Note that $\sum_{e \in B} \mathsf{left}(\delta(e)) = 1$ holds in the definition of this case, and that $\sum_{s'' \in S \setminus \{s'\}} \mathsf{left}(\delta(s, s'')) < 1$ holds for all $(s, s') \in E$ from the definition of edges. Combining these two facts allows us to conclude that $\mathsf{left}(\delta(e)) > 0$ for all $e \in B$. Then $\mathbf{a}(s') = \mathsf{left}(\delta(s, s')) + f(s') > 0$ for all $s' \in T_B$, establishing that $\mathsf{support}(\mathbf{a}) \supseteq T_B$.

**Case** $\sum_{e \in B} \mathsf{left}(\delta(e)) < 1$ and $\sum_{e \in B} \mathsf{right}(\delta(e)) = 1$. We let $f(s') = \mathsf{right}(\delta(s, s')) - \mathsf{left}(\delta(s, s'))$ for all $s' \in T_B$, and let $f(s') = 0$ for all $s' \in S \setminus T_B$. Hence $\mathsf{left}(\delta(s, s')) + f(s') = \mathsf{right}(\delta(s, s'))$ for all $s' \in T_B$, and $\mathsf{left}(\delta(s, s')) + f(s') = \mathsf{left}(\delta(s, s'))$ for all $s' \in S \setminus T_B$. First we establish that the choice of $f$ above yields an assignment by showing that conditions (a) and (b) in the definition of $f$ are satisfied. First consider condition (a). The antecedent of condition $(\tilde{4})$ is satisfied for this case, hence $\delta(e)$ is right-closed for all $e \in B$, *i.e.*, $\mathsf{right}(\delta(s, s')) \in \delta(s, s')$ for all $s' \in T_B$. For $s' \in S \setminus T_B$, because $f(s') = 0$, we need to show that $\mathsf{left}(\delta(s, s')) \in \delta(s, s')$, *i.e.*, that $\delta(s, s')$ is left-closed. Recalling that $T_{E(s) \setminus B}$ and $S_{\neg E(s)}$ form a partition of $S \setminus T_B$, we have two sub-cases based on whether $s'$ belongs to $T_{E(s) \setminus B}$ or to $S_{\neg E(s)}$. For $s' \in T_{E(s) \setminus B}$, given that $E(s) \setminus B \subseteq E^{[0, \cdot\rangle}$ from the realisability of $B$, we have that $\delta(s, s')$ is left-closed. For $s' \in S_{\neg E(s)}$, from the definition of edges, we either have $\mathsf{right}(\delta(s, s')) = 0$ or $\sum_{s'' \in S \setminus \{s'\}} \mathsf{left}(\delta(s, s'')) \geq 1$. For $\mathsf{right}(\delta(s, s')) = 0$, it must be the case that $\delta(s, s') = [0, 0]$, which is left-closed. For $\sum_{s'' \in S \setminus \{s'\}} \mathsf{left}(\delta(s, s'')) \geq 1$, given that well-formedness condition (1) specifies that $\sum_{s'' \in S} \mathsf{left}(\delta(s, s'')) \leq 1$, we must have $\sum_{s'' \in S} \mathsf{left}(\delta(s, s'')) = 1$. Then well-formedness condition (2) specifies that $\delta(s, s')$ is left-closed. Hence we have established condition (a) in the definition of $f$. Next we turn our attention to condition (b) in the definition of $f$. Recall that $T_B$, $T_{E(s) \setminus B}$ and $S_{\neg E(s)}$ form a partition of $S$. First consider

$T_B$. Noting that $\sum_{e\in B}\mathsf{right}(\delta(e)) = 1$, we have $\sum_{s'\in T_B}\mathsf{right}(\delta(s,s')) = 1$. Then, from the choice of $f$, we have $\sum_{s'\in T_B}\mathsf{left}(\delta(s,s')) + f(s') = 1$. Now consider $T_{E(s)\setminus B}$. Recall that, for $s' \in T_{E(s)\setminus B}$, we have chosen $f(s') = 0$ and established that $\mathsf{left}(\delta(s,s')) = 0$ (because $\delta(s,s') = [0,0]$ above, and hence $\sum_{s'\in T_{E(s)\setminus B}}\mathsf{left}(\delta(s,s')) + f(s') = 0$. Finally, for $s' \in S_{\neg E(s)}$, we either have $\mathsf{right}(\delta(s,s')) = 0$, in which case $\mathsf{left}(\delta(s,s')) = 0$, or $\sum_{s''\in S\setminus\{s'\}}\mathsf{left}(\delta(s,s'')) \geq 1$, in which case the well-formedness condition (1) specifying that $\sum_{s''\in S}\mathsf{left}(\delta(s,s'')) \leq 1$ establishes that $\mathsf{left}(\delta(s,s')) = 0$. Given that $f(s') = 0$ for all $s' \in S\setminus T_B$ and $S_{\neg E(s)} \subseteq S\setminus T_B$, we have $\sum_{s'\in S_{\neg E(s)}}\mathsf{left}(\delta(s,s')) + f(s') = 0$. From the fact that $T_B$, $T_{E(s)\setminus B}$ and $S_{\neg E(s)}$ form a partition of $S$, we then conclude that:

$$\sum_{s'\in S}\mathsf{left}(\delta(s,s')) + f(s') = \sum_{s'\in T_B}\mathsf{right}(\delta(s,s')) = 1.$$

Hence we have established condition (b) in the definition of $f$, and therefore the choice of $f$ yields an assignment.

It remains to show that $\mathsf{support}(\mathbf{a}) = T_B$. From the reasoning in the previous two paragraphs, we have established that $\mathsf{left}(\delta(s,s')) + f(s') = \mathsf{right}(\delta(s,s'))$ for all $s' \in T_B$, and $\mathsf{left}(\delta(s,s')) + f(s') = 0$ for all $s' \in S\setminus T_B$. Hence $\mathsf{support}(\mathbf{a}) \subseteq T_B$. From the fact that $\mathsf{right}(\delta(s,s')) > 0$ for $s' \in S$ such that $(s,s') \in E(s)$, and from $B \subseteq E(s)$, we have that $\mathsf{right}(\delta(s,s')) > 0$ for all $s' \in T_B$. Hence $\mathsf{support}(\mathbf{a}) \supseteq T_B$. We then conclude that $\mathsf{support}(\mathbf{a}) = T_B$.

**Case** $\sum_{e\in B}\mathsf{left}(\delta(e)) < 1 < \sum_{e\in B}\mathsf{right}(\delta(e))$. For each $s' \in T_B$, we let:

$$f(s') = (1 - \sum_{s''\in T_B}\mathsf{left}(\delta(s,s''))) \cdot \frac{\mathsf{width}(\delta(s,s'))}{\sum_{s''\in T_B}\mathsf{width}(\delta(s,s''))},$$

and let $f(s') = 0$ for each $s' \in S\setminus T_B$.

First we show that this definition of $f$ satisfies condition (a), *i.e.*, $\mathsf{left}(\delta(s,s')) + f(s') \in \delta(s,s')$ for all $s' \in S$. In the sub-case in which $\mathsf{left}(\delta(s,s')) = \mathsf{right}(\delta(s,s'))$ (*i.e.*, $\delta(s,s')$ is the degenerate interval $[\mathsf{left}(\delta(s,s')), \mathsf{right}(\delta(s,s'))]$), we have $f(s') = 0$, because either $s' \in S\setminus T_B$, or $s' \in T_B$ and $f(s') = 0$ is a consequence of the fact that $\mathsf{width}(\delta(s,s')) = 0$. As in the proof of Proposition 1, we then conclude that $\mathsf{left}(\delta(s,s')) + f(s') \in \delta(s,s')$. Now consider the sub-case in which $\mathsf{left}(\delta(s,s')) < \mathsf{right}(\delta(s,s'))$. Recall again that $T_B$, $T_{E(s)\setminus B}$ and $S_{\neg E(s)}$ form a partition of $S$. If $s' \in T_{E(s)\setminus B}$ or $s' \in S_{\neg E(s)}$, we conclude, following identical reasoning used in the previous case, that $\mathsf{left}(\delta(s,s')) + f(s') \in \delta(s,s')$. On the other hand, if $s' \in T_B$, we proceed in a similar manner to that of the analogous case of Proposition 1. In order to show $\mathsf{left}(\delta(s,s')) + f(s') \in \delta(s,s')$, we will establish that $0 < f(s') < \mathsf{width}(\delta(s,s'))$. The fact that $f(s') > 0$ is a consequence of the following three facts: $1 - \sum_{e\in B}\mathsf{left}(\delta(e)) > 0$ (from the definition of the case), $\mathsf{width}(\delta(s,s')) > 0$ (given that we are considering the sub-case of $\mathsf{left}(\delta(s,s')) < \mathsf{right}(\delta(s,s')))$, and $\sum_{s''\in T_B}\mathsf{width}(\delta(s,s'')) > 0$ (from the previous fact and from the fact that $\mathsf{width}(\delta(s,s'')) \geq 0$ for all $s'' \in S$). We now show that $f(s') < \mathsf{width}(\delta(s,s'))$. From the definition of $f$, this requires showing that $\frac{1-\sum_{s''\in T_B}\mathsf{left}(\delta(s,s''))}{\sum_{s''\in T_B}\mathsf{width}(\delta(s,s''))} < 1$. Given that we are considering the case in which $1 < \sum_{e\in B}\mathsf{right}(\delta(e))$, *i.e.*, $1 < \sum_{s''\in T_B}\mathsf{left}(\delta(s,s'')) + \mathsf{width}(\delta(s,s''))$, we can conclude

that $1 - \sum_{s'' \in T_B} \mathsf{left}(\delta(s,s'')) < \sum_{s'' \in T_B} \mathsf{width}(\delta(s,s''))$ and hence $\frac{1 - \sum_{s'' \in T_B} \mathsf{left}(\delta(s,s''))}{\sum_{s'' \in T_B} \mathsf{width}(\delta(s,s''))} < 1$. Thus we have established that the choice of $f(s')$ satisfies $\mathsf{left}(\delta(s,s')) + f(s') \in \delta(s,s')$ for $s' \in T_B$. We now proceed to show that $f$ satisfies condition (b), $i.e.$, $\sum_{s' \in S} \mathsf{left}(\delta(s,s')) + f(s') = 1$. Recall that $T_B$, $T_{E(s) \setminus B}$ and $S_{\neg E(s)}$ form a partition of $S$, and that $f(s') = 0$ for each $s' \in S \setminus T_B$. Following similar reasoning to that of the previous case and of the proof of Proposition 1, we show that $\mathsf{left}(\delta(s,s')) = 0$, and hence $\mathsf{left}(\delta(s,s')) + f(s') = 0$, for each $s' \in S \setminus T_B$. For $s' \in T_{E(s) \setminus B}$, we have $\mathsf{left}(\delta(s,s')) = 0$ from the fact that $E(s) \setminus B \subseteq E^{[0,\cdot\rangle}$ because $B$ is realisable. For $s' \in S_{\neg E(s)}$, we have $\mathsf{left}(\delta(s,s')) = 0$ by the definition of edges: either $\mathsf{right}(\delta(s,s')) = 0$ and hence $\mathsf{left}(\delta(s,s')) = 0$, or the combination of $\sum_{s'' \in S \setminus \{s'\}} \mathsf{left}(\delta(s,s'')) \geq 1$ and $\sum_{s'' \in S} \mathsf{left}(\delta(s,s'')) \leq 1$ (well-formedness condition (1)) establishes that $\mathsf{left}(\delta(s,s')) = 0$. Given that $\mathsf{left}(\delta(s,s')) + f(s') = 0$, for each $s' \in S \setminus T_B$, our aim reduces to showing that $\sum_{s' \in T_B} \mathsf{left}(\delta(s,s')) + f(s') = 1$. We proceed as for the analogous case in Proposition 1:

$$\sum_{s' \in T_B} \mathsf{left}(\delta(s,s')) + f(s')$$

$$= \sum_{s' \in T_B} \mathsf{left}(\delta(s,s')) + \sum_{s' \in T_B} f(s')$$

$$= \sum_{s' \in T_B} \mathsf{left}(\delta(s,s')) + \sum_{s' \in T_B} (1 - \sum_{s'' \in T_B} \mathsf{left}(\delta(s,s''))) \cdot \frac{\mathsf{width}(\delta(s,s'))}{\sum_{s'' \in T_B} \mathsf{width}(\delta(s,s''))}$$

$$= \sum_{s' \in T_B} \mathsf{left}(\delta(s,s')) + (1 - \sum_{s'' \in T_B} \mathsf{left}(\delta(s,s''))) \cdot \frac{\sum_{s' \in T_B} \mathsf{width}(\delta(s,s'))}{\sum_{s'' \in T_B} \mathsf{width}(\delta(s,s''))}$$

$$= \sum_{s' \in T_B} \mathsf{left}(\delta(s,s')) + (1 - \sum_{s'' \in T_B} \mathsf{left}(\delta(s,s'')))$$

$$= 1.$$

Next, we show that $\mathsf{support}(\mathbf{a}) = T_B$. The fact that $\mathsf{support}(\mathbf{a}) \subseteq T_B$ follows from $\mathsf{left}(\delta(s,s')) + f(s') = 0$, for each $s' \in S \setminus T_B$, which was established in the previous paragraph. In order to establish that $\mathsf{support}(\mathbf{a}) \supseteq T_B$, we observe the following facts. For $s' \in T_B$, we have $(s,s') \in B \subseteq E(s)$, and hence $\mathsf{right}(\delta(s,s')) > 0$ from the definition of edges. Then if $\mathsf{width}(\delta(s,s')) = 0$, from $\mathsf{left}(\delta(s,s')) = \mathsf{right}(\delta(s,s'))$ we have $\mathsf{left}(\delta(s,s')) > 0$. Instead, if $\mathsf{width}(\delta(s,s')) > 0$, we have $f(s') > 0$ from the choice of $f$. Hence, for each $s' \in T_B$, we have $\mathsf{left}(\delta(s,s')) + f(s') > 0$, from which we obtain $\mathsf{support}(\mathbf{a}) \supseteq T_B$. We conclude that $\mathsf{support}(\mathbf{a}) = T_B$.

This concludes the ($\Rightarrow$) direction of the proof.

($\Leftarrow$) Let $\mathbf{a}$ be an assignment for state $s$ such that $\mathsf{support}(\mathbf{a}) = T_B$. Our aim is to show that $B$ is valid, $i.e.$, that $B$ is large and realisable. From the definition of assignments, we have $\mathbf{a}(s') \in \delta(s,s')$, and hence $\mathbf{a}(s') \leq \mathsf{right}(\delta(s,s'))$ for each state $s' \in S$. From this fact, and given that $\mathbf{a}$ is a distribution ($i.e.$, sums to 1), we have:

$$\sum_{(s,s') \in B} \mathsf{right}(\delta(s,s')) = \sum_{s' \in \mathsf{support}(\mathbf{a})} \mathsf{right}(\delta(s,s')) \geq \sum_{s' \in \mathsf{support}(\mathbf{a})} \mathbf{a}(s') = 1.$$

In the case of $\sum_{(s,s')\in B}\mathsf{right}(\delta(s,s')) > 1$, the fact that $B$ is large follows immediately from the definition of largeness. In the case of $\sum_{(s,s')\in B}\mathsf{right}(\delta(s,s')) = 1$, then $\mathbf{a}(s') = \mathsf{right}(\delta(s,s'))$ (*i.e.*, $\mathbf{a}(s')$ must be equal to the right endpoint of $\delta(s,s')$) for each $(s,s') \in B$. Hence $B \subseteq E^{\langle\cdot,\cdot]}$, and as a consequence $B$ is large. The fact that $\{(s,s')|s' \in \mathsf{support}(\mathbf{a})\} = B$ implies that $\mathbf{a}(s') = 0$ for all $(s,s') \in E(s) \setminus B$. Hence $E(s) \setminus B \subseteq E^{[0,\cdot)}$, and therefore $B$ is realisable. Because $B$ is large and realisable, $B$ is valid. ∎

A consequence of Lemma 1, together with Proposition 1 and the fact that we consider only well-formed IMCs, is that there exists at least one valid subset of outgoing edges from each state.

For each state $s \in S$, we let $Valid(s) = \{B \subseteq E(s)|B \text{ is valid}\}$. Note that $|Valid(s)| = 2^{|E(s)|} - 1$ in the worst case (when all edges in $E(s)$ are associated with intervals $[0,1]$). Given a valid set $B \in Valid(s)$, we let $ValidAssign(B)$ be the set of assignments $\mathbf{a}$ for $s$ that witness Lemma 1, *i.e.*, all assignments $\mathbf{a}$ for $s$ such that $\{(s,s')|s' \in \mathsf{support}(\mathbf{a})\} = B$. Let $Valid = \bigcup_{s \in S} Valid(s)$ be the set of valid sets of the IMC. A *witness assignment function* $w : Valid \to \mathsf{Dist}(S)$ assigns to each valid set $B \in Valid$ an assignment from $ValidAssign(B)$.

**Example 1** For the state $s_1$ of the IMC $\mathscr{O}_2$ of Fig. 2, the valid edge sets are $B_1 = \{(s_1,s_0),(s_1,s_1),(s_1,s_2)\}$ *and* $B_2 = \{(s_1,s_0),(s_1,s_2)\}$, reflecting the intuition that the edge $(s_1,s_1)$ can be assigned (exactly) probability 0. Note that modifying the IMC so that the right endpoint of $(s_1,s_0)$ is reduced to $0.7$ would result in $B_1$ being the only valid set associated with $s_1$; the set $B_2$ would not be large, because it is not possible to assign probability 0 to edge $(s_1,s_1)$ and total probability of 1 to edges $(s_1,s_0)$ and $(s_1,s_2)$. An example of a witness assignment function $w$ for state $s_1$ of $\mathscr{O}_2$ is $w(B_1)(s_0) = 0.7$, $w(B_1)(s_1) = 0.12$ and $w(B_1)(s_2) = 0.18$, and $w(B_2)(s_0) = 0.8$ and $w(B_2)(s_2) = 0.2$. □

The *qualitative MDP abstraction of* $\mathscr{O}$ *with respect to witness assignment function* $w$ is the MDP $[\mathscr{O}]_w = (S, \Delta_w)$, where $\Delta_w$ is defined by $\Delta_w(s) = \{w(B)|B \in Valid(s)\}$ for each state $s \in S$.

## QUALITATIVE REACHABILITY: UMC SEMANTICS

Qualitative reachability problems can be classified into four categories, depending on whether we ask that the probability of reaching the target set $T$ is 0 or 1 for some or for all ways of assigning probabilities to intervals. For the UMC semantics of IMC $\mathscr{O} = (S, \delta)$, state $s \in S$, quantifier $\mathsf{Q} \in \{\forall, \exists\}$ and probability $\lambda \in \{0, 1\}$, the $(s, \mathsf{Q}, \lambda)$-*reachability problem for the UMC semantics of* $\mathscr{O}$ asks whether

$$\mathsf{Q}\mathscr{D} \in [\mathscr{O}]_\mathsf{U} . \mathsf{Pr}_s^{\mathscr{D}}(\mathsf{Reach}(T)) = \lambda$$

holds. To solve the $(s, \mathsf{Q}, \lambda)$-reachability problem for the UMC semantics of $\mathscr{O}$, we compute the set $S_\mathsf{Q}^{\lambda, \mathsf{U}} = \{s' \in S|\mathsf{Q}\mathscr{D} \in [\mathscr{O}]_\mathsf{U} . \mathsf{Pr}_{s'}^{\mathscr{D}}(\mathsf{Reach}(T)) = \lambda\}$, then check whether $s \in S_\mathsf{Q}^{\lambda, \mathsf{U}}$. Portions of this text were previously published as part of a preprint (*Sproston, 2018b*).

**Theorem 1** *Let* $\mathsf{Q} \in \{\forall, \exists\}$ *and* $\lambda \in \{0, 1\}$. *The set* $S_\mathsf{Q}^{\lambda, \mathsf{U}}$ *can be computed in polynomial time in the size of the IMC.* □

A straightforward corollary of Theorem 1 is that qualitative reachability problems for the UMC semantics of IMCs are in P. The remainder of this section is dedicated to showing Theorem 1 for each of the combinations of $Q \in \{\forall, \exists\}$ and $\lambda \in \{0, 1\}$.

## Computation of $S_{\forall}^{0,U}$

The case for $S_{\forall}^{0,U}$ is straightforward. We compute the complement of $S_{\forall}^{0,U}$, namely the set of states for which there exists a DTMC in the UMC semantics such that $T$ is reached with positive probability, written formally as $S \setminus S_{\forall}^{0,U} = \{s \in S | \exists \mathscr{D} \in [\mathscr{O}]_U . \Pr_s^{\mathscr{D}}(\mathsf{Reach}(T)) > 0\}$. The computation of this set reduces to reachability on the graph of the IMC according to the following lemma.

**Lemma 2** *Let $s \in S$. There exists $\mathscr{D} \in [\mathscr{O}]_U$ such that $\Pr_s^{\mathscr{D}}(\mathsf{Reach}(T)) > 0$ if and only if there exists a finite path $r \in Paths_*^{\mathscr{O}}(s)$ such that $last(r) \in T$.*

**Proof** ($\Rightarrow$) Let $\mathscr{D} = (S, \mathbf{P}) \in [\mathscr{O}]_U$ be a DTMC such that $\Pr_s^{\mathscr{D}}(\mathsf{Reach}(T)) > 0$. From $\Pr_s^{\mathscr{D}}(\mathsf{Reach}(T)) > 0$, there exists at least one finite path $s_0 s_1 \cdots s_n \in Paths_*^{\mathscr{D}}(s)$ of $\mathscr{D}$ such that $s_n \in T$. For each $i < n$, we have that $\mathbf{P}(s_i, s_{i+1}) > 0$. From the definition of the UMC semantics, we have that $\mathbf{P}(s_i, \cdot)$ is an assignment for $s_i$ such that $\mathbf{P}(s_i, \cdot)(s_{i+1}) > 0$, from which we then obtain that $(s_i, s_{i+1}) \in E$ by Proposition 1. Repeating this reasoning for each $i < n$, we have that $s_0 s_1 \cdots s_n \in Paths_*^{\mathscr{O}}(s)$. Recalling that $s_n \in T$, this direction of the proof is completed.

($\Leftarrow$) Let $s_0 s_1 \cdots s_n \in Paths_*^{\mathscr{O}}(s)$ be a finite path of $\mathscr{O}$ such that $s_n \in T$, which we assume w.l.o.g. does not contain any cycle. By the definition of finite paths of $\mathscr{O}$, we have $(s_i, s_{i+1}) \in E$ for each $i < n$. From Proposition 1, for each $i < n$, there exists an assignment $\mathbf{a}$ for state $s_i$ such that $\mathbf{a}(s_{i+1}) > 0$. In turn, this means that there exists a DTMC $\mathscr{D} = (S, \mathbf{P}) \in [\mathscr{O}]_U$ such that $\mathbf{P}(s_i, s_{i+1}) > 0$ for each $i < n$ (for any state $t \in S \setminus \{s_0, s_1, \ldots, s_n\}$ not visited along the path, the distribution $\mathbf{P}(t, \cdot)$ can be defined in an arbitrary way). Given that $s_n \in T$, we obtain $\Pr_s^{\mathscr{D}}(\mathsf{Reach}(T)) > 0$. ∎

Hence the set $S_{\forall}^{0,U}$ is equal to the complement of the set of states from which there exists a path reaching $T$ in the graph of the IMC. Given that the graph of the IMC and the latter set of states can be computed in polynomial time, we conclude that $S_{\forall}^{0,U}$ can be computed in polynomial time.

## Computation of $S_{\exists}^{0,U}$

We show that $S_{\exists}^{0,U}$ can be obtained by computing, in the qualitative MDP abstraction $[\mathscr{O}]_w = (S, \Delta_w)$ of $\mathscr{O}$ with respect to some (arbitrary) witness assignment function $w$, the set of states for which there exists a scheduler such that $T$ is reached with probability 0.

To establish the correctness of this approach, we show a more general result: for $\lambda \in \{0, 1\}$, the set of states of $[\mathscr{O}]_w$ for which there exists a scheduler such that $T$ is reached with probability $\lambda$ is equal to the set of states of $\mathscr{O}$ for which there exists a DTMC in $[\mathscr{O}]_U$ such that $T$ is reached with probability $\lambda$. The case in which $\lambda = 0$ will be used subsequently for the computation of $S_{\exists}^{0,U}$, whereas the case in which $\lambda = 1$ will be used later in the article for the computation of $S_{\exists}^{1,U}$.

**Lemma 3** *Let $s \in S$ and $\lambda \in \{0, 1\}$. There exists $\mathscr{D} \in [\mathscr{O}]_U$ such that $\Pr_s^{\mathscr{D}}(\mathsf{Reach}(T)) = \lambda$ if and only if there exists a scheduler $\sigma \in \Sigma^{[\mathscr{O}]_w}$ such that $\Pr_s^{\sigma}(\mathsf{Reach}(T)) = \lambda$.*

**Proof** ($\Rightarrow$) Let $\mathscr{D} = (S, \mathbf{P})$ be a DTMC such that $\mathscr{D} \in [\mathscr{O}]_U$ and $\mathrm{Pr}_s^{\mathscr{D}}(\mathsf{Reach}(T)) = \lambda$. We define the (memoryless) scheduler $\sigma_{\mathscr{D}} \in \Sigma^{[\mathscr{O}]_w}$ of $[\mathscr{O}]_w$ in the following way. Consider a state $s' \in S$, and let $B_{s'} = \{(s', s'') \in E | \mathbf{P}(s', s'') > 0\}$ be the set of edges with source $s'$ assigned positive probability by $\mathscr{D}$. Note that $B_{s'} \in Valid(s')$: this follows from the fact that $\mathbf{P}(s', \cdot)$ is an assignment for $s'$ and by Lemma 1. Then, for any finite path $r \in Paths_*^{[\mathscr{O}]_w}(s)$ ending in state $s'$ (that is, $last(r) = s'$), we let $\sigma_{\mathscr{D}}(r) = \{w(B_{s'}) \mapsto 1\}$, i.e., $\sigma_{\mathscr{D}}$ chooses (with probability 1) the distribution that corresponds to the witness assignment function applied to the edge set $B_{s'}$ (this is possible because $B_{s'} \in Valid(s')$ and $\Delta_w(s') = \{w(B) | B \in Valid(s')\}$ by definition). From the fact that $\sigma_{\mathscr{D}}$ is memoryless, we can obtain the folded DTMC $\tilde{\mathscr{D}}^{\sigma_{\mathscr{D}}} = (S, \tilde{\mathbf{P}})$ of $\sigma_{\mathscr{D}}$. Now observe that the DTMC $\mathscr{D}$ and the folded DTMC $\tilde{\mathscr{D}}^{\sigma_{\mathscr{D}}}$ are *graph equivalent* in the following sense: for any $s', s'' \in S$, we have $\mathbf{P}(s', s'') > 0$ if and only if $\tilde{\mathbf{P}}(s', s'') > 0$ (which follows from the fact that, for all states $s' \in S$, we have $\{(s', s'') \in S \times S | \mathbf{P}(s', s'') > 0\} = B_{s'} = \{(s', s'') | s'' \in \mathsf{support}(w(B_{s'}))\} = \{(s', s'') \in S \times S | \tilde{\mathbf{P}}(s', s'') > 0\}$). Then, from (*Chatterjee, Sen & Henzinger, 2008*, Lemma 2), which specifies that the sets of states satisfying any given qualitative $\omega$-regular (and hence also reachability) property in graph equivalent DTMCs are identical, we have that $\mathrm{Pr}_s^{\mathscr{D}}(\mathsf{Reach}(T)) = \lambda$ implies $\mathrm{Pr}_s^{\sigma_{\mathscr{D}}}(\mathsf{Reach}(T)) = \lambda$.

($\Leftarrow$) Assume that there exists a scheduler $\sigma \in \Sigma^{[\mathscr{O}]_w}$ of $[\mathscr{O}]_w$ such that $\mathrm{Pr}_s^{\sigma}(\mathsf{Reach}(T)) = \lambda$. Given that $[\mathscr{O}]_w$ is a finite MDP, from standard results (*de Alfaro, 1997*; *Baier & Katoen, 2008*), we can assume that $\sigma$ is (a) pure (that is, for all finite paths $r \in Paths_*^{[\mathscr{O}]_w}$, we have that $\sigma(r) = \{\mu \mapsto 1\}$ for some $\mu \in \Delta_w(last(r))$) and (b) memoryless. Given (a) and (b), we can write $\sigma$ as a mapping $\sigma : S \to \bigcup_{s' \in S} \Delta_w(s')$. Then consider the DTMC $\mathscr{D} = (S, \mathbf{P})$, where $\mathbf{P}(s', \cdot) = \sigma(s')$ for all states $s' \in S$. Note that, for all states $s' \in S$, we have that $\sigma(s')$ is an assignment, from the following two facts: (1) $\sigma$ chooses a distribution $w(B)$ for some $B \in Valid(s')$, and (2) $w$ is a witness assignment function, i.e., $w(B)$ is an assignment. Given that $\sigma(s')$ is an assignment for all $s' \in S$, hence $\mathbf{P}(s', \cdot)$ is an assignment for all $s' \in S$, and therefore $\mathscr{D} \in [\mathscr{O}]_U$. The folded DTMC $\tilde{\mathscr{D}}^{\sigma}$ and the DTMC $\mathscr{D}$ are graph equivalent. Hence, given that $\mathrm{Pr}_s^{\sigma}(\mathsf{Reach}(T)) = \lambda$, we have $\mathrm{Pr}_s^{\mathscr{D}}(\mathsf{Reach}(T)) = \lambda$. ∎

In particular, Lemma 3 allows us to reduce the problem of computing $S_{\exists}^{0, U}$ to that of computing the set $S_{\exists}^{0, w} = \{s \in S | \exists \sigma \in \Sigma^{[\mathscr{O}]_w} . \mathrm{Pr}_s^{\sigma}(\mathsf{Reach}(T)) = 0\}$ on $[\mathscr{O}]_w$ for an arbitrary witness assignment function $w$. As in the case of standard finite MDP techniques (see *Forejt et al. (2011)*), we proceed by computing the *complement* of this set, i.e., we compute the set $S \setminus S_{\exists}^{0, w} = \{s \in S | \forall \sigma \in \Sigma^{[\mathscr{O}]_w} . \mathrm{Pr}_s^{\sigma}(\mathsf{Reach}(T)) > 0\}$. For a state set $X \subseteq S$, let $\mathsf{CPre}(X)$ be the set of states for which there exists a distribution such that all states assigned positive probability by the distribution are in $X$, defined formally as $\mathsf{CPre}(X) = \{s \in S | \exists \mu \in \Delta_w(s) . \mathsf{support}(\mu) \subseteq X\}$. Furthermore, let $\overline{\mathsf{CPre}}(X)$ be set of states such that all available distributions make a transition to $X$ with positive probability, defined formally as $\overline{\mathsf{CPre}}(X) = \{s \in S | \forall \mu \in \Delta_w(s) . \mathsf{support}(\mu) \cap X \neq \emptyset\}$. Note that $\overline{\mathsf{CPre}}$ is the dual of the $\mathsf{CPre}$ operator, i.e., $\overline{\mathsf{CPre}}(X) = S \setminus \mathsf{CPre}(S \setminus X)$. The standard algorithm for computing the set of states of a finite MDP for which all schedulers result in reaching a set $T$ of target states with probability strictly greater than 0 operates in the following way: starting from $X_0 = T$, we let $X_{i+1} = X_i \cup \overline{\mathsf{CPre}}(X_i)$ for progressively larger values of $i \geq 0$,

until we reach a fixpoint (that is, until we obtain $X_{i^*+1} = X_{i^*}$ for some $i^*$). However, a direct application of this algorithm to $[\mathscr{O}]_w$ results in an exponential-time algorithm, given that the size of the transition function $\Delta_w$ of $[\mathscr{O}]_w$ is in general exponential in the size of $\mathscr{O}$. For this reason, we propose an algorithm that operates directly on the IMC $\mathscr{O}$, without requiring the explicit construction of $[\mathscr{O}]_w$. We proceed by establishing that CPre can be implemented in polynomial time in the size of $\mathscr{O}$.

**Lemma 4** *Let* $s \in S$ *and* $X \subseteq S$. *Then:*

- $s \in \mathsf{CPre}(X)$ *if and only if* $E(s, X)$ *is valid;*
- *the set* $\mathsf{CPre}(X)$ *can be computed in polynomial time in the size of* $\mathscr{O}$.

**Proof** The proof of the first part of Lemma 4 relies on Lemma 1 and the observation that for any $B, B' \subseteq E(s, X)$ such that $B \subseteq B'$, if $E(s) \setminus B \subseteq E^{[0, \cdot)}$ and $B$ is large, then $E(s) \setminus B' \subseteq E^{[0, \cdot)}$ and $B'$ is large (intuitively, the greater the subset of $E(s, X)$, the easier it is to satisfy realisability and largeness). Formally, the observation holds because $E(s) \setminus B' \subseteq E(s) \setminus B$, and, in the case of $B \subset B'$, for $e \in B' \setminus B$, given that $\mathsf{right}(\delta(e)) > 0$ (from the definition of edges) and $\sum_{e' \in B} \mathsf{right}(\delta(e')) \geq 1$ (because $B$ is large), we must have $\sum_{e' \in B'} \mathsf{right}(\delta(e')) > 1$ (that is, condition (a) in the definition of largeness holds for $B'$, regardless of whether condition (a) or (b) in the definition of largeness holds for $B$). Hence if there exists $B \subseteq E(s, X)$ such that $E(s) \setminus B \subseteq E^{[0, \cdot)}$ and $B$ is large, then $E(s) \setminus E(s, X) \subseteq E^{[0, \cdot)}$ and $E(s, X)$ is large. We now show formally the first part of Lemma 4:

$s \in \mathsf{CPre}(X)$

| | | |
|---|---|---|
| $\Leftrightarrow$ | $\exists \mu \in \Delta_w(s).\mathsf{support}(\mu) \subseteq X$ | (definition of CPre) |
| $\Leftrightarrow$ | $\exists B \subseteq E(s).B \subseteq S \times X \wedge B \in Valid(s)$ | (Lemma 1, definition of $[\mathscr{O}]_w$) |
| $\Leftrightarrow$ | $\exists B \subseteq E(s).B \subseteq S \times X \wedge E(s) \setminus B \subseteq E^{[0, \cdot)} \wedge B$ is large | (definition of $Valid(s)$) |
| $\Leftrightarrow$ | $E(s, X) \subseteq S \times X \wedge E(s) \setminus E(s, X) \subseteq E^{[0, \cdot)} \wedge E(s, X)$ is large | (observation above) |
| $\Leftrightarrow$ | $E(s) \setminus E(s, X) \subseteq E^{[0, \cdot)} \wedge E(s, X)$ is large | (definition of $E(s, X)$) |
| $\Leftrightarrow$ | $E(s, X)$ is valid | (definition of validity). |

The second part of Lemma 4 follows from the first part of the lemma and the fact that checking validity of $E(s, X)$ (that is, checking that $E(s) \setminus E(s, X) \subseteq E^{[0, \cdot)}$ and $E(s, X)$ is large) can be done in polynomial time in the size of $\mathscr{O}$. ∎

**Example 2** In Fig. 3, we illustrate the intuition underlying Lemma 4, making reference to an IMC fragment (left) and its corresponding qualitative MDP abstraction fragment (right; note that probabilities are not represented for the qualitative MDP abstraction to avoid clutter). Our aim is to determine whether $s_0$ belongs to $\mathsf{CPre}(X)$ for $X = \{s_1, s_2\}$. According to Lemma 4, determining whether $s_0$ belongs to $\mathsf{CPre}(X)$ is equivalent to determining whether $E(s_0, X)$ is valid. The edge set $E(s_0, X)$ is large (because $\mathsf{right}(\delta(s_0, s_1)) + \mathsf{right}(\delta(s_0, s_2)) = 0.5 + 0.6 = 1.1 > 1$) and realisable (because $(s_0, s_3) \in E^{[0, \cdot)}$). Hence, by reasoning on the IMC, we can determine the existence of a distribution from $s_0$ in the qualitative MDP abstraction (indicated by the black square) such that the distribution assigns positive probability only to states in $X$. For completeness, we also note that state $s_0$ is associated not just with the valid edge set $\{(s_0, s_1), (s_0, s_2)\}$, but also with the valid edge sets $\{(s_0, s_1), (s_0, s_3)\}$ and $\{(s_0, s_1), (s_0, s_2), (s_0, s_3)\}$, which correspond to the distributions illustrated with the grey squares with the support sets $\{s_1, s_3\}$ *and* $\{s_1, s_2, s_3\}$, respectively, in the qualitative MDP abstraction. □

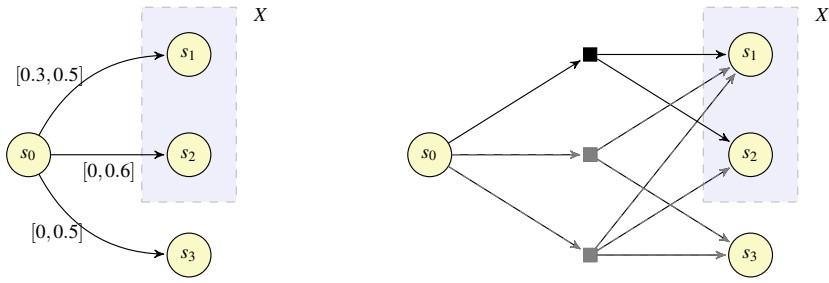

**Figure 3  Example fragments of an IMC (left) and the corresponding qualitative MDP abstraction (right).**

Based on Lemma 4, instead of constructing and analysing the qualitative MDP abstraction $[\mathscr{O}]_w$, we propose computing directly on $\mathscr{O}$ the sets $X_0 = T$ and $X_{i+1} = X_i \cup \overline{\mathsf{CPre}}(X_i)$ for increasing indices $i$ until a fixpoint is reached. Given that a fixpoint must be reached within $|S|$ steps, and the fact that Lemma 4 specifies that $\mathsf{CPre}(S \setminus X_i)$, and hence $\overline{\mathsf{CPre}}(X_i)$, can be done in polynomial time in the size of $\mathscr{O}$, we have that the set $\{s \in S | \forall \sigma \in \Sigma^{[\mathscr{O}]_w} . \mathsf{Pr}_s^\sigma(\mathsf{Reach}(T)) > 0\}$ can be computed in polynomial time in the size of $\mathscr{O}$. The complement of this set is equal to $S_\exists^{0,\mathsf{U}}$, as established by Lemma 3, and hence we can compute $S_\exists^{0,\mathsf{U}}$ in polynomial time in the size of $\mathscr{O}$.

## Computation of $S_\exists^{1,\mathsf{U}}$

We proceed in a manner analogous to that for the case of $S_\exists^{0,\mathsf{U}}$. First note that, by Lemma 3, the set $S_\exists^{1,\mathsf{U}}$ is equal to the set of states of $[\mathscr{O}]_w$ for which there exists a scheduler that results in $T$ being reached with probability 1, i.e., $S_\exists^{1,\mathsf{U}} = \{s \in S | \exists \sigma \in \Sigma^{[\mathscr{O}]_w} . \mathsf{Pr}_s^\sigma(\mathsf{Reach}(T)) = 1\}$. Hence our aim is to compute $\{s \in S | \exists \sigma \in \Sigma^{[\mathscr{O}]_w} . \mathsf{Pr}_s^\sigma(\mathsf{Reach}(T)) = 1\}$ on $[\mathscr{O}]_w$. We recall the standard algorithm for the computation of this set on finite MDPs (*de Alfaro, 1997*; *de Alfaro, 1999*). Given state sets $X, Y \subseteq S$, we let

$$\mathsf{APre}(Y, X) = \{s \in S | \exists \mu \in \Delta_w(s) . \mathsf{support}(\mu) \subseteq Y \wedge \mathsf{support}(\mu) \cap X \neq \emptyset\}.$$

That is, $\mathsf{APre}(Y, X)$ is the set of states for which there exists a distribution such that (a) all states assigned positive probability by the distribution are in $Y$ and (b) there exists a state in $X$ that is assigned positive probability by the distribution. The standard algorithm for computing the set of states for which there exists a scheduler that results in $T$ being reached with probability 1 proceeds as follows. First we set $Y_0 = S$ and $X_0^0 = T$. Then sequence $X_0^0, X_1^0, \ldots$ is computed by letting $X_{i_0+1}^0 = X_{i_0}^0 \cup \mathsf{APre}(Y_0, X_{i_0}^0)$ for progressively larger indices $i_0 \geq 0$ until a fixpoint is obtained, that is, until we obtain $X_{i_0^*+1}^0 = X_{i_0^*}^0$ for some $i_0^*$. Next we let $Y_1 = X_{i_0^*}^0$, $X_0^1 = T$ and compute $X_{i_1+1}^1 = X_{i_1}^1 \cup \mathsf{APre}(Y_1, X_{i_1}^1)$ for larger $i_1 \geq 0$ until a fixpoint $X_{i_1^*}^1$ is obtained. Then we let $Y_2 = X_{i_1^*}^1$ and $X_0^2 = T$, and repeat the process. We terminate the algorithm when a fixpoint is reached in the sequence $Y_0, Y_1, \ldots$.[3] The algorithm requires at most $|S|^2$ calls to $\mathsf{APre}$. In an analogous manner to

---

[3] Readers familiar with $\mu$-calculus will observe that the algorithm can be expressed using the term $\nu Y . \mu X (T \cup \mathsf{APre}(Y, X))$ (*de Alfaro, 1999*).

CPre in the case of $S_\exists^{0,U}$, we show that APre can be characterised by efficiently checkable conditions on $\mathcal{O}$.

**Lemma 5** *Let* $s \in S$ *and let* $X, Y \subseteq S$. *Then:*

- $s \in \mathsf{APre}(Y, X)$ *if and only if* $E(s, X \cap Y) \neq \emptyset$, *and* $E(s, Y)$ *is valid;*
- *the set* $\mathsf{APre}(Y, X)$ *can be computed in polynomial time in the size of* $\mathcal{O}$.

**Proof** The intuition underlying Lemma 5 is similar to that of Lemma 4. In particular, observe that if there exists $B \subseteq E(s, Y)$ such that $B \cap (S \times X) \neq \emptyset$, $E(s) \setminus B \subseteq E^{[0,\cdot)}$ and $B$ is large, then $E(s, Y) \cap (S \times X) \neq \emptyset$, $E(s) \setminus E(s, Y) \subseteq E^{[0,\cdot)}$ and $E(s, Y)$ is large. We first show formally the first part of the lemma:

$s \in \mathsf{APre}(Y, X)$

| | | |
|---|---|---|
| $\Leftrightarrow$ | $\exists \mu \in \Delta_w(s).\mathsf{support}(\mu) \subseteq Y \wedge \mathsf{support}(\mu) \cap X \neq \emptyset$ | (definition of APre) |
| $\Leftrightarrow$ | $\exists B \in E(s).B \subseteq S \times Y \wedge B \cap (S \times X) \neq \emptyset \wedge B \in \mathit{Valid}(s)$ | (Lemma 1, definition of $[\mathcal{O}]_w$) |
| $\Leftrightarrow$ | $\exists B \in E(s).B \subseteq S \times Y \wedge B \cap (S \times X) \neq \emptyset \wedge E(s) \setminus B \subseteq E^{[0,\cdot)} \wedge B$ is large | (definition of $\mathit{Valid}(s)$) |
| $\Leftrightarrow$ | $E(s, Y) \subseteq S \times Y \wedge E(s, Y) \cap (S \times X) \neq \emptyset \wedge E(s) \setminus E(s, Y) \subseteq E^{[0,\cdot)} \wedge E(s, Y)$ is large | (observation above) |
| $\Leftrightarrow$ | $E(s, X \cap Y) \neq \emptyset \wedge E(s, S \setminus Y) \subseteq E^{[0,\cdot)} \wedge E(s, Y)$ is large | (definition of $E(s, Y)$) |
| $\Leftrightarrow$ | $E(s, X \cap Y) \neq \emptyset \wedge E(s, Y)$ is valid | (definition of validity). |

The second part of the lemma follows from the first part combined with the fact that checking whether $E(s, X \cap Y) \neq \emptyset$ and $E(s, Y)$ is valid can be done in polynomial time in the size of $\mathcal{O}$.

Hence we obtain an overall polynomial-time algorithm for computing $\{s \in S | \exists \sigma \in \Sigma^{[\mathcal{O}]_w}.\mathrm{Pr}_s^\sigma(\mathsf{Reach}(T)) = 1\}$ which, from Lemma 3, equals $S_\exists^{1,U}$.

## Computation of $S_\forall^{1,U}$

We recall the standard algorithm for determining the set of states for which all schedulers reach a target set with probability 1 on a finite MDP (see *Forejt et al. (2011)*): from the set of states of the MDP, we first remove states from which the target state can be reached with probability 0 for some scheduler, then successively remove states for which it is possible to reach a previously removed state with positive probability. For each of the remaining states, all schedulers result in the target set being reached with probability 1.

We propose an algorithm for IMCs that is inspired by this standard algorithm for finite MDPs. Our aim is to compute the complement of $S_\forall^{1,U}$, *i.e.*, the state set $S \setminus S_\forall^{1,U} = \{s \in S | \exists \mathscr{D} \in [\mathcal{O}]_U.\mathrm{Pr}_s^\mathscr{D}(\mathsf{Reach}(T)) < 1\}$.

**Lemma 6** *Let* $s \in S$. *There exists* $\mathscr{D} \in [\mathcal{O}]_U$ *such that* $\mathrm{Pr}_s^\mathscr{D}(\mathsf{Reach}(T)) < 1$ *if and only if there exists a path* $r \in \mathit{Paths}_*^\mathcal{O}(s)$ *such that* $\mathit{last}(r) \in S_\exists^{0,U}$.

**Proof** $(\Rightarrow)$ Let $\mathscr{D} = (S, \mathbf{P}) \in [\mathcal{O}]_U$ be a DTMC such that $\mathrm{Pr}_s^\mathscr{D}(\mathsf{Reach}(T)) < 1$. A bottom strongly connected component (BSCC) $V \subseteq S$ of $\mathscr{D}$ is a strongly connected component of the graph of $\mathscr{D}$ (that is, the graph $(S, \{(s', s'') | \mathbf{P}(s', s'') > 0\})$) such that there is no outgoing edge from $V$ (that is, for all states $s' \in V$ and $s'' \in S$, if $\mathbf{P}(s', s'') > 0$ then $s'' \in V$). Let $\mathscr{V} \subseteq 2^S$ be the set of BSCCs of $\mathscr{D}$. Given that $\mathscr{D}$ is a finite DTMC, by standard results for finite DTMCs (see, for example, *Baier & Katoen (2008)*), we have that BSCCs are reached with probability 1, *i.e.*, $\mathrm{Pr}_s^\mathscr{D}(\mathsf{Reach}(\bigcup_{V \in \mathscr{V}} V)) = 1$, and that once a BSCC is entered all of its states are visited with probability 1. Hence $\mathrm{Pr}_s^\mathscr{D}(\mathsf{Reach}(T)) < 1$ implies that there exists some BSCC $V \in \mathscr{V}$ such that $V \cap T = \emptyset$ and $\mathrm{Pr}_s^\mathscr{D}(\mathsf{Reach}(V)) > 0$. Next we repeat the reasoning of Lemma 2 to show that there exists a finite path $s_0 s_1 \cdots s_n \in \mathit{Paths}_*^\mathcal{O}(s)$ such that

$s_n \in V$. We now show that $V \subseteq S_\exists^{0,\mathrm{U}}$: for any state $s' \in V$, the fact that $V \cap T = \emptyset$ and from the fact that BSCCs do not feature outgoing edges, we have that $\mathrm{Pr}_{s'}^{\mathscr{D}}(\mathsf{Reach}(T)) = 0$, and hence $s' \in S_\exists^{0,\mathrm{U}}$. Therefore, for the finite path $s_0 s_1 \cdots s_n \in Paths_*^{\mathscr{O}}(s)$, we have $s_n \in S_\exists^{0,\mathrm{U}}$, which completes this direction of the proof.

($\Leftarrow$) Let $s_0 s_1 \cdots s_n \in Paths_*^{\mathscr{O}}(s)$ be a finite path of $\mathscr{O}$ such that $s_n \in S_\exists^{0,\mathrm{U}}$. We assume w.l.o.g. that $s_0 s_1 \cdots s_n$ does not contain any cycle, and that $s_i \notin S_\exists^{0,\mathrm{U}}$ for all $i < n$. As in the proof of Lemma 2, we can use these facts to conclude that there exists a DTMC $\mathscr{D} = (S, \mathbf{P}) \in [\mathscr{O}]_\mathrm{U}$ such that $\mathbf{P}(s_i, s_{i+1}) > 0$ for each $i < n$. Furthermore, from the definition of $S_\exists^{0,\mathrm{U}}$, for any state $s' \in S_\exists^{0,\mathrm{U}}$, there exists a DTMC $\mathscr{D}^{s'} = (S, \mathbf{P}_{s'}) \in [\mathscr{O}]_\mathrm{U}$ such that $\mathrm{Pr}_{s'}^{\mathscr{D}^{s'}}(\mathsf{Reach}(T)) = 0$. In particular, from $s_n \in S_\exists^{0,\mathrm{U}}$, we have $\mathrm{Pr}_{s_n}^{\mathscr{D}^{s_n}}(\mathsf{Reach}(T)) = 0$. Note that all states on all paths in $Paths_*^{\mathscr{D}^{s_n}}(s_n)$ are in $S_\exists^{0,\mathrm{U}}$ (if this was not the case, that is there exists a path in $Paths_*^{\mathscr{D}^{s_n}}(s_n)$ featuring a state $s'' \notin S_\exists^{0,\mathrm{U}}$, then we must also have $s_n \notin S_\exists^{0,\mathrm{U}}$). The fact that the set $\{s_0, s_1, \ldots, s_{n-1}\}$ (*i.e.*, the states before $s_n$ along the finite path) and the set $S_\exists^{0,\mathrm{U}}$ are disjoint means that we can combine the DTMCs $\mathscr{D}$ (along the finite path) and $\mathscr{D}^{s_n}$ (within $S_\exists^{0,\mathrm{U}}$) to obtain a DTMC $\mathscr{D}' = (S, \mathbf{P}')$; formally, $\mathbf{P}'(s_i, \cdot) = \mathbf{P}(s_i, \cdot)$ for $i < n$, $\mathbf{P}'(s', \cdot) = \mathbf{P}s_n(s', \cdot)$ for $s' \in S_\exists^{0,\mathrm{U}}$, and $\mathbf{P}'(s', \cdot)$ can be defined by arbitrary assignments for all other states, *i.e.*, for $s' \in S \setminus (\{s_0, s_1, \ldots, s_{n-1}\} \cup S_\exists^{0,\mathrm{U}})$. Given that $\mathscr{D}, \mathscr{D}^{s_n} \in [\mathscr{O}]_\mathrm{U}$, we have that $\mathscr{D}' \in [\mathscr{O}]_\mathrm{U}$. Furthermore, in $\mathscr{D}'$, from $s$ there exists a finite path to states $S_\exists^{0,\mathrm{U}}$ in which the DTMC is subsequently confined, and (by the trivial fact that $S_\exists^{0,\mathrm{U}} \subseteq S \setminus T$) therefore $\mathrm{Pr}_s^{\mathscr{D}'}(\mathsf{Reach}(T)) < 1$, completing this direction of the proof. ∎

Hence the set $S_\forall^{1,\mathrm{U}}$ can be computed by taking the complement of the set of states for which there exists a path to $S_\exists^{0,\mathrm{U}}$ in the graph of $\mathscr{O}$. Given that $S_\exists^{0,\mathrm{U}}$, and the set of states reaching $S_\exists^{0,\mathrm{U}}$, can be computed in polynomial time, we have obtained a polynomial-time algorithm for computing $S_\forall^{1,\mathrm{U}}$. Together with the cases for $S_\forall^{0,\mathrm{U}}$, $S_\exists^{0,\mathrm{U}}$ and $S_\exists^{1,\mathrm{U}}$, this establishes Theorem 1.

## QUALITATIVE REACHABILITY: IMDP SEMANTICS

We now focus on the IMDP semantics of IMC $\mathscr{O} = (S, \delta)$. Qualitative reachability problems in this setting take a similar form to that seen for the UMC semantics, with the key difference being the quantification over schedulers in $\Sigma^{[\mathscr{O}]_\mathrm{I}}$ in the IMDP semantics, rather than over DTMCs in $[\mathscr{O}]_\mathrm{U}$ in the UMC semantics. For state $s \in S$, quantifier $\mathsf{Q} \in \{\forall, \exists\}$ and probability $\lambda \in \{0, 1\}$, the $(s, \mathsf{Q}, \lambda)$-*reachability problem for the IMDP semantics of* $\mathscr{O}$ asks whether

$$\mathsf{Q}\sigma \in \Sigma^{[\mathscr{O}]_\mathrm{I}} . \mathrm{Pr}_s^\sigma(\mathsf{Reach}(T)) = \lambda$$

holds. To solve the $(s, \mathsf{Q}, \lambda)$-reachability problem for the IMDP semantics of $\mathscr{O}$, we compute the set $S_\mathsf{Q}^{\lambda,\mathrm{I}} = \{s' \in S | \mathsf{Q}\sigma \in \Sigma^{[\mathscr{O}]_\mathrm{I}} . \mathrm{Pr}_{s'}^\sigma(\mathsf{Reach}(T)) = \lambda\}$, then check whether $s \in S_\mathsf{Q}^{\lambda,\mathrm{I}}$. Portions of this text were previously published as part of a preprint (*Sproston, 2018b*). This section will be dedicated to showing the following result. We note that the cases for $S_\forall^{0,\mathrm{I}}$, $S_\exists^{0,\mathrm{I}}$ and $S_\exists^{1,\mathrm{I}}$ proceed in a manner similar to the UMC case (using either

graph reachability or reasoning based on the qualitative MDP abstraction); instead the case for $S_\forall^{1,\mathrm{I}}$ requires substantially different techniques.

**Theorem 2** *Let* $\mathsf{Q} \in \{\forall, \exists\}$ *and* $\lambda \in \{0, 1\}$. *The set* $S_\mathsf{Q}^{\lambda,\mathrm{I}}$ *can be computed in polynomial time in the size of the IMC.*

A corollary of Theorem 2 is the membership in P of qualitative reachability problems for the IMDP semantics of IMCs. In the rest of this section, we establish Theorem 2 for $\mathsf{Q} \in \{\forall, \exists\}$ and $\lambda \in \{0, 1\}$.

## Computation of $S_\forall^{0,\mathrm{I}}$

As in the case of UMCs, the computation of $S_\forall^{0,\mathrm{I}}$ relies on a straightforward reachability analysis on the graph of the IMC $\mathscr{O}$ to obtain the complement of $S_\forall^{0,\mathrm{I}}$. The correctness of this approach is based on the following lemma.

**Lemma 7** *Let* $s \in S$. *There exists* $\sigma \in \Sigma^{[\mathscr{O}]_\mathrm{I}}$ *such that* $\mathrm{Pr}_s^\sigma(\mathsf{Reach}(T)) > 0$ *if and only if there exists a path* $r \in Paths_*^\mathscr{O}(s)$ *such that* $last(r) \in T$. $\qquad\square$

**Proof** $(\Rightarrow)$ Let $\sigma \in \Sigma^{[\mathscr{O}]_\mathrm{I}}$ be a scheduler of $[\mathscr{O}]_\mathrm{I}$ such that $\mathrm{Pr}_s^\sigma(\mathsf{Reach}(T)) > 0$. Recall that the DTMC $\mathscr{D}_s^\sigma$ is defined as $(Paths_*^\sigma(s), \mathbf{P}_s^\sigma)$: the state set of $\mathscr{D}_s^\sigma$ comprises the finite paths resulting from choices of $\sigma$ from state $s$, and, for finite paths $r, r' \in Paths_*^\sigma(s)$, we have $\mathbf{P}_s^\sigma(r, r') > 0$ if $r' = r\mu s'$ such that $\mu \in \Delta(last(r))$, $\sigma(\mu) > 0$ and $\mu(s') > 0$, otherwise $\mathbf{P}_s^\sigma(r, r') = 0$. Furthermore, we say that a path $\rho$ of $\mathscr{D}_s^\sigma$ is in $\mathsf{Reach}(T)$ if there exists a finite path in $Paths_*^\sigma(s)$, with last state in $T$, that is visited along $\rho$. For a path $\rho$ of $\mathscr{D}_s^\sigma$, we can obtain a path $\rho'$ of $\mathscr{O}$ simply by extracting the sequence of the final states for all finite prefixes of $\rho$: that is, from the path $s_0(s_0\mu_0 s_1)(s_0\mu_0 s_1\mu_1 s_2)\cdots$ of $\mathscr{D}_s^\sigma$ we can obtain the path $s_0 s_1 s_2 \cdots$ of $\mathscr{O}$, because $\mu_i(s_{i+1}) > 0$ implies that $(s_i, s_{i+1}) \in E$ for all $i \in \mathbb{N}$ (from the definition of $[\mathscr{O}]_\mathrm{I}$ and Lemma 1). Observe that $\rho \in \mathsf{Reach}(T)$ implies that $\rho' \in \mathsf{Reach}(T)$. To conclude, $\mathrm{Pr}_s^\sigma(\mathsf{Reach}(T)) > 0$ implies that there exists $\rho \in Paths_\infty^{\mathscr{D}_s^\sigma}(s)$ such that $\rho \in \mathsf{Reach}(T)$, which in turn implies that there exists $\rho' \in Paths_\infty^\mathscr{O}(s)$ such that $\rho' \in \mathsf{Reach}(T)$, i.e., there exists a finite prefix $r$ of $\rho'$ (and therefore $r \in Paths_*^\mathscr{O}(s)$) such that $last(r) \in T$.

$(\Leftarrow)$ Let $s_0 s_1 \cdots s_n \in Paths_*^\mathscr{O}(s)$ be a finite path of $\mathscr{O}$ such that $s_n \in T$. By the definition of finite paths of $\mathscr{O}$, we have $(s_i, s_{i+1}) \in E$ for each $i < n$, which, by Lemma 1, implies in turn that there exists an assignment $\mathbf{a}_i$ to $s_i$ such that $\mathbf{a}_i(s_{i+1}) > 0$. We consider a scheduler $\sigma \in \Sigma^{[\mathscr{O}]_\mathrm{I}}$ of $[\mathscr{O}]_\mathrm{I}$ defined in the following way: for each $i < n$, for all paths $r \in Paths_*^{[\mathscr{O}]_\mathrm{I}}(s)$ such that $last(r) = s_i$, then we let $\sigma(r) = \mathbf{a}_i$ ($\sigma$ can be defined in an arbitrary manner for all other finite paths). Observe that the finite path $r = s_0(s_0\mathbf{a}_0 s_1)(s_0\mathbf{a}_0 s_1\mathbf{a}_1 s_2)\cdots(s_0\mathbf{a}_0\cdots\mathbf{a}_{n-1}s_n)$ is a finite path of $\mathscr{D}_s^\sigma$ such that all infinite paths of $\mathscr{D}_s^\sigma$ that have $r$ as a prefix are in $\mathsf{Reach}(T)$, and hence $\mathrm{Pr}_s^\sigma(\mathsf{Reach}(T)) > 0$. $\qquad\blacksquare$

Therefore, to obtain $S_\forall^{0,\mathrm{I}}$, we compute the state set
$S \setminus S_\forall^{0,\mathrm{I}} = \{s \in S | \exists \sigma \in \Sigma^{[\mathscr{O}]_\mathrm{I}}. \mathrm{Pr}_s^\sigma(\mathsf{Reach}(T)) > 0\}$, which reduces to reachability on the graph of the IMC according to Lemma 7, and then take the complement of the resulting set.

## Computation of $S_\exists^{0,\mathrm{I}}$ and $S_\exists^{1,\mathrm{I}}$

In the following we fix an arbitrary witness assignment function $w$ of $\mathscr{O}$. Lemma 8 establishes that $S_\exists^{0,\mathrm{I}}$ (respectively, $S_\exists^{1,\mathrm{I}}$) equals the set of states of the qualitative MDP abstraction $[\mathscr{O}]_w$ with respect to $w$ for which there exists some scheduler such that $T$ is reached with probability 0 (respectively, probability 1).

**Lemma 8** *Let $s \in S$ and $\lambda \in \{0, 1\}$. There exists a scheduler $\sigma \in \Sigma^{[\mathscr{O}]_\mathrm{I}}$ such that $\mathrm{Pr}_s^\sigma(\mathsf{Reach}(T)) = \lambda$ if and only if there exists a scheduler $\sigma' \in \Sigma^{[\mathscr{O}]_w}$ such that $\mathrm{Pr}_s^{\sigma'}(\mathsf{Reach}(T)) = \lambda$.* □

**Proof** We first deal comprehensively with the case of $\lambda = 0$, and then consider the case of $\lambda = 1$.

Case: $\lambda = 0$. ($\Rightarrow$) Let $\sigma \in \Sigma^{[\mathscr{O}]_\mathrm{I}}$ be a scheduler of $[\mathscr{O}]_\mathrm{I}$ such that $\mathrm{Pr}_s^\sigma(\mathsf{Reach}(T)) = 0$. We show how we can define a scheduler $\mathbf{f}(\sigma) \in \Sigma^{[\mathscr{O}]_w}$ such that $\mathrm{Pr}_s^{\mathbf{f}(\sigma)}(\mathsf{Reach}(T)) = 0$.

For any finite path $r \in Paths_*^{[\mathscr{O}]_\mathrm{I}}(s)$, let $B_r^\sigma$ be the set of edges assigned positive probability by $\sigma$ after $r$. Formally:

$$B_r^\sigma = \{(s, s') \in E \mid s = last(r) \wedge \exists \mu \in \Delta(s).(\sigma(r)(\mu) > 0 \wedge \mu(s') > 0)\}.$$

We now define the partial function $\mathbf{g} : Paths_*^{[\mathscr{O}]_w}(s) \to Paths_*^\sigma(s)$, which associates with a finite path of $[\mathscr{O}]_w$ some finite path of $[\mathscr{O}]_\mathrm{I}$ (more precisely, of $\sigma$) that features the same sequence of states. For $r = s_0 \mu_0 s_1 \mu_1 \cdots \mu_{n-1} s_n \in Paths_*^{[\mathscr{O}]_w}(s)$, let $\mathbf{g}(r) = s_0' \mu_0' s_1' \mu_1' \cdots \mu_{n-1}' s_n'$ such that:

- $s_i' = s_i$ for all $i \le n$;
- $\mathbf{g}(s_0 \mu_0 s_1 \mu_1 \cdots \mu_{i-1} s_i) = s_0' \mu_0' s_1' \mu_1' \cdots \mu_{i-1}' s_i'$ for all $i < n$;
- $\sigma(\mathbf{g}(s_0 \mu_0 s_1 \mu_1 \cdots \mu_{i-1} s_i))(\mu_i') > 0$ and $\mu_i'(s_{i+1}') > 0$ for all $i < n$.

The first and third conditions, and $s_0 = s$, ensure that $\mathbf{g}(r) \in Paths_*^\sigma(s)$; if these conditions do not hold, then $\mathbf{g}(r)$ is undefined. The second condition ensures that $\mathbf{g}$ maps paths of $Paths_*^{[\mathscr{O}]_w}(s)$ to paths of $Paths_*^\sigma(s)$ in a consistent manner (for example, the condition ensures that, if there are two finite paths $r_1$ and $r_2$ of $[\mathscr{O}]_w$ with a common prefix of length $i$, then $\mathbf{g}(r_1)$ and $\mathbf{g}(r_2)$ will also have a common prefix of length $i$). We are now in a position to define the scheduler $\mathbf{f}(\sigma)$ of $[\mathscr{O}]_w$: let $\mathbf{f}(\sigma)(r) = \{w(B_{\mathbf{g}(r)}^\sigma) \mapsto 1\}$ for each finite path $r \in Paths_\infty^{[\mathscr{O}]_w}(s)$ (recall that $w(B_{\mathbf{g}(r)}^\sigma)$ is an assignment that witnesses Lemma 1 for $B_{\mathbf{g}(r)}^\sigma$).

Next we need to show that $\mathrm{Pr}_s^{\mathbf{f}(\sigma)}(\mathsf{Reach}(T)) = 0$. Note that $\mathrm{Pr}_s^\sigma(\mathsf{Reach}(T)) = 0$ if and only if all paths $\rho \in Paths_\infty^\sigma(s)$ are such that $\rho \notin \mathsf{Reach}(T)$, and similarly $\mathrm{Pr}_s^{\mathbf{f}(\sigma)}(\mathsf{Reach}(T)) = 0$ if and only if all paths $\rho \in Paths_\infty^{\mathbf{f}(\sigma)}(s)$ are such that $\rho \notin \mathsf{Reach}(T)$. Hence we show that the existence of a path $\rho \in Paths_\infty^{\mathbf{f}(\sigma)}(s)$ such that $\rho \in \mathsf{Reach}(T)$ implies the existence of a path $\rho' \in Paths_\infty^\sigma(s)$ such that $\rho' \in \mathsf{Reach}(T)$. Assuming the existence of some $\rho \in Paths_\infty^{\mathbf{f}(\sigma)}(s)$ such that $\rho \in \mathsf{Reach}(T)$, we will identify a path $\rho' \in Paths_\infty^\sigma(s)$ that visits the same states as $\rho$, and hence is such that $\rho' \in \mathsf{Reach}(T)$. Writing $\rho = s_0 \mu_0 s_1 \mu_1 \cdots$, and letting $r_i$ be the $i$th prefix of $\rho$ (i.e., $r_i = s_0 \mu_0 s_1 \mu_1 \cdots \mu_{i-1} s_i$), we now show the existence of $\rho'$ by induction, by considering prefixes $r_{i'}$ of $\rho'$ of increasing length.

*(Base case.)* Let $r_0' = s_0$.

*(Inductive step.)* Assume that we have constructed the finite path $r_i' = s_0 \mu_0' s_1 \mu_1' \cdots \mu_{i-1}' s_i$. We show how to extend $r_i'$ with one transition to obtain the finite path $r_{i+1}'$. The transition $s_i \mu_i s_{i+1}$ along $\rho$ implies that $(s_i, s_{i+1}) \in B_{\mathbf{g}(r_i)}^{\sigma}$. This fact then implies that there exists some $\mu' \in \Delta(s_i)$ such that $\sigma(\mathbf{g}(r_i))(\mu') > 0$ and $\mu'(s_{i+1}) > 0$. We then let $r_{i+1}' = r_i' \mu' s_{i+1}$.

From this construction of $\rho'$, we can see that $\rho' \in Paths_\infty^\sigma(s)$ (because $\rho'$ follows the definition of a path of scheduler $\sigma$, and because the choice of $\mu'$ after $r_i'$ must be consistent with the choices made along all prefixes of $r_i'$). Furthermore, we have $\rho' \in \mathsf{Reach}(T)$ (because $\rho$ and $\rho'$ feature the same states, and $\rho \in \mathsf{Reach}(T)$). Hence we have shown that the existence of $\rho \in Paths_\infty^{\mathbf{f}(\sigma)}(s)$ such that $\rho \in \mathsf{Reach}(T)$ implies the existence of $\rho' \in Paths_\infty^\sigma(s)$ such that $\rho' \in \mathsf{Reach}(T)$. This fact means that $\Pr_s^\sigma(\mathsf{Reach}(T)) = 0$ implies $\Pr_s^{\mathbf{f}(\sigma)}(\mathsf{Reach}(T)) = 0$, concluding this direction of the proof.

($\Leftarrow$) Let $\sigma \in \Sigma^{[\mathscr{O}]_w}$ such that $\Pr_s^\sigma(\mathsf{Reach}(T)) = 0$. Given that $[\mathscr{O}]_w$ is a finite MDP, we can assume that $\sigma$ is memoryless and pure. We now define $\sigma' \in \Sigma^{[\mathscr{O}]_I}$ and show that $\Pr_s^{\sigma'}(\mathsf{Reach}(T)) = 0$. For a finite path $r \in Paths_*^{[\mathscr{O}]_I}(s)$, let $\sigma'(r) = \sigma(last(r))$ (recall that, for all states $s' \in S$, $\sigma(s') = \{w(B) \mapsto 1\}$ for some $B \in Valid(s')$, where $w(B)$ is an assignment for $s'$, hence $\sigma'$ is well defined). Then we have that the DTMCs $\mathscr{D}_s^\sigma$ and $\mathscr{D}_s^{\sigma'}$ are identical, and hence $\Pr_s^\sigma(\mathsf{Reach}(T)) = 0$ implies that $\Pr_s^{\sigma'}(\mathsf{Reach}(T)) = 0$.

Hence the part of the proof for $\lambda = 0$ is concluded.

Case: $\lambda = 1$. ($\Rightarrow$) Let $\sigma \in \Sigma^{[\mathscr{O}]_I}$ be such that $\Pr_s^\sigma(\mathsf{Reach}(T)) = 1$. We use the construction of the scheduler $\mathbf{f}(\sigma)$ of $[\mathscr{O}]_I$ from the case of $S_\exists^{0,I}$ (*i.e.*, the case $\lambda = 0$ of this proof), and show that $\Pr_s^{\mathbf{f}(\sigma)}(\mathsf{Reach}(T)) = 1$. We show that $\Pr_s^\sigma(\mathsf{Reach}(T)) = 1$ implies $\Pr_s^{\mathbf{f}(\sigma)}(\mathsf{Reach}(T)) = 1$ by showing the contrapositive, *i.e.*, $\Pr_s^{\mathbf{f}(\sigma)}(\mathsf{Reach}(T)) < 1$ implies $\Pr_s^\sigma(\mathsf{Reach}(T)) < 1$. To show this property, we use *end components* (*de Alfaro, 1997*). An end component of finite MDP $[\mathscr{O}]_w = (S, \Delta_w)$ is a pair $(C, D)$ where $C \subseteq S$ and $D : C \to 2^{\mathsf{Dist}(S)}$ is such that (1) $\emptyset \neq D(s') \subseteq \Delta_w(s')$ for all $s' \in C$, (2) $\mathsf{support}(\mu) \subseteq C$ for all $s' \in C$ and $\mu \in D(s')$, and (3) the graph $(C, \{(s', s'') \in C \times C | \exists \mu \in D(s') . \mu(s'') > 0\})$ is strongly connected. Let $\mathscr{E}$ be the set of end components of $[\mathscr{O}]_w$. For an end component $(C, D)$, let $sa(C, D) = \{(s', \mu) | s' \in C \wedge \mu \in D(s')\}$ be the set of state-action pairs associated with $(C, D)$. For a path $\rho \in Paths_\infty^{[\mathscr{O}]_w}(s)$, we let $inf(\rho) = \{(s, \mu) | s$ and $\mu$ appear infinitely often along $\rho\}$. The fundamental theorem of end components (*de Alfaro, 1997*) specifies that, for any scheduler $\sigma' \in \Sigma^{[\mathscr{O}]_w}$, we have that $\Pr_s^{\sigma'}(\{\rho | \exists (C, D) \in \mathscr{E} . inf(\rho) = sa(C, D)\}) = 1$.

Now, given that $[\mathscr{O}]_w$ is a finite MDP, from $\Pr_s^{\mathbf{f}(\sigma)}(\mathsf{Reach}(T)) < 1$ and the fact that states in $T$ are absorbing, there exists an end component $(C, D) \in \mathscr{E}$ such that $C \cap T = \emptyset$ and $\Pr_s^{\mathbf{f}(\sigma)}(\{\rho | inf(\rho) = sa(C, D)\}) > 0$. We observe the following property: for any finite path $r \in Paths_*^{\mathbf{f}(\sigma)}(s)$, we have that $\{(last(r), s') | w(B_{\mathbf{g}(r)}^\sigma)(s') > 0\} = B_{\mathbf{g}(r)}^\sigma$, due to the fact that $w(B_{\mathbf{g}(r)}^\sigma)$ is a valid assignment for $B_{\mathbf{g}(r)}^\sigma$. From the definition of $B_{\mathbf{g}(r)}^\sigma$, we can see that $\sigma$ assigns positive probability only to those distributions that assign positive probability to states that are targets of edges in $B_{\mathbf{g}(r)}^\sigma$. Putting these two facts together, we conclude that, for any set $X \subseteq S$, we have $\mathsf{support}(w(B_{\mathbf{g}(r)}^\sigma)) \subseteq X$ if and only if $\{s' \in S | \exists \mu \in \Delta(last(\mathbf{g}(\rho))) . (\sigma(\mathbf{g}(r))(\mu) > 0 \wedge \mu(s') > 0)\} \subseteq X$. This means that the existence of a

finite path $r \in Paths_*^{\mathbf{f}(\sigma)}(s)$ such that all suffixes of $r$ generated by $\mathbf{f}(\sigma)$ visit only states in $C$ implies that all suffixes of $\mathbf{g}(r)$ (which we recall is a finite path of $\sigma$, *i.e.*, $\mathbf{g}(r) \in Paths_*^{\sigma}(s)$) generated by $\sigma$ visit only states in $C$. Given that $\Pr_s^{\mathbf{f}(\sigma)}(\{\rho | inf(\rho) = sa(C,D)\}) > 0$, such a finite path in $Paths_*^{\mathbf{f}(\sigma)}(s)$ exists. Then, given that there exists a finite path in $Paths_*^{\sigma}(s)$ such that all suffixes generated by $\sigma$ visit only states in $C$, and from the fact that $C \cap T = \emptyset$, we have that $\Pr_s^{\sigma}(\mathsf{Reach}(T)) < 1$. Hence we have shown that $\Pr_s^{\mathbf{f}(\sigma)}(\mathsf{Reach}(T)) < 1$ implies that $\Pr_s^{\sigma}(\mathsf{Reach}(T)) < 1$. This means that $\Pr_s^{\sigma}(\mathsf{Reach}(T)) = 1$ implies that $\Pr_s^{\mathbf{f}(\sigma)}(\mathsf{Reach}(T)) = 1$. Hence, for $\sigma \in \Sigma^{[\mathscr{O}]_\mathsf{I}}$ such that $\Pr_s^{\sigma}(\mathsf{Reach}(T)) = 1$ there exists $\sigma' \in \Sigma^{[\mathscr{O}]_w}$ such that $\Pr_s^{\sigma'}(\mathsf{Reach}(T)) = 1$.

($\Leftarrow$) Let $\sigma \in \Sigma^{[\mathscr{O}]_w}$ be such that $\Pr_s^{\sigma}(\mathsf{Reach}(T)) = 1$. As in the case of the analogous direction of the case for $\lambda = 0$, we can show the existence of $\sigma' \in \Sigma^{[\mathscr{O}]_\mathsf{I}}$ such that $\mathscr{D}_s^{\sigma}$ and $\mathscr{D}_s^{\sigma'}$ are identical, and hence $\Pr_s^{\sigma}(\mathsf{Reach}(T)) = 1$ implies that $\Pr_s^{\sigma'}(\mathsf{Reach}(T)) = 1$. ■

Given that we have shown in Section 'Qualitative Reachability: UMC semantics' that the set of states of the qualitative MDP abstraction $[\mathscr{O}]_w$ for which there exists some scheduler such that $T$ is reached with probability 0 (respectively, probability 1) can be computed in polynomial time in the size of $\mathscr{O}$, we obtain polynomial-time algorithms for computing $S_\exists^{0,\mathsf{I}}$ (respectively, $S_\exists^{1,\mathsf{I}}$).

The following corollary summarises the relationship between the state sets satisfying analogous reachability problems in the UMC and IMDP semantics, and is obtained by combining Lemma 2, Lemma 3, Lemma 7 and Lemma 8.

**Corollary 1** $S_\forall^{0,\mathsf{U}} = S_\forall^{0,\mathsf{I}}$, $S_\exists^{0,\mathsf{U}} = S_\exists^{0,\mathsf{I}}$ and $S_\exists^{1,\mathsf{U}} = S_\exists^{1,\mathsf{I}}$. □

## Computation of $S_\forall^{1,\mathsf{I}}$

This case is notably different from the other three cases for the IMDP semantics, because schedulers that are *not* memoryless may influence whether a state is included in $S_\forall^{1,\mathsf{I}}$. In particular, we recall the example of the IMC of Fig. 1: as explained in Section 'Introduction', we have $s_0 \notin S_\forall^{1,\mathsf{I}}$. In contrast, for the UMC semantics, we have $s_0 \in S_\forall^{1,\mathsf{U}}$, and $s_0$ would be in $S_\forall^{1,\mathsf{I}}$ if we restricted the IMDP semantics to memoryless (actually finite-memory) schedulers. For this reason, a qualitative MDP abstraction is not useful for computing $S_\forall^{1,\mathsf{I}}$, because it is based on the use of witness assignment functions that assign *constant* probabilities to sets of edges available from states: on repeated visits to a state, the (finite) set of available distributions remains the same in a qualitative MDP abstraction. Therefore we require alternative analysis methods that are not based on the qualitative MDP abstraction. In this section we introduce an alternative notion of end components, defined solely in terms of states of the IMC, which characterise situations in which the IMC can confine its behaviour to certain state sets with positive probability in the IMDP semantics: for example, the IMC of Fig. 1 can confine itself to state $s_0$ with positive probability in the IMDP semantics. The key characteristic of these end components is that the total probability assigned to edges that have a source state in the component but a target state outside of the component can be made to be arbitrarily small (note that

such edges must have an interval with a left endpoint of 0). We now define formally our alternative notion of end components.

**Definition 3 (ILECs):**  *A set $C \subseteq S$ of states is an* IMC-level end component *(ILEC) if, for each state $s \in C$, we have (1) $E^{\langle +, \cdot \rangle}(s, S \setminus C) = \emptyset$, (2) $\sum_{e \in E(s,C)} \mathsf{right}(\delta(e)) \geq 1$, and (3) the sub-graph $(C, E \cap (C \times C))$ is strongly connected.* □

**Example 3**  *In the IMC $\mathcal{O}_1$ of Fig. 1, the set $\{s_0\}$ is an ILEC: for condition (1), the edge $(s_0, s_1)$ (the only edge in $E(s_0, S \setminus \{s_0\})$) is not in $E^{\langle +, \cdot \rangle}$, and, for condition (2), we have $\mathsf{right}(\delta(s_0, s_0)) = 1$. In the IMC $\mathcal{O}_2$ of Fig. 2, the set $\{s_0, s_1\}$ is an ILEC: for condition (1), the only edge leaving $\{s_0, s_1\}$ has 0 as its left endpoint, i.e., $\delta(s_1, s_2) = (0, 0.2]$, hence $E^{\langle +, \cdot \rangle}(s_0, \{s_2\}) = E^{\langle +, \cdot \rangle}(s_1, \{s_2\}) = \emptyset$; for condition (2), we have $\mathsf{right}(\delta(s_0, s_0)) + \mathsf{right}(\delta(s_0, s_1)) = 1.6 \geq 1$ and $\mathsf{right}(\delta(s_1, s_0)) + \mathsf{right}(\delta(s_1, s_1)) = 1.3 \geq 1$. In both cases, the identified sets clearly induce strongly connected subgraphs, thus satisfying condition (3).* □

**Remark 2** Both conditions (1) and (2) are necessary to ensure that the probability of leaving $C$ in one step can be made arbitrarily small. Consider an IMC with state $s \in C$ such that $E(s, C) = \{e_1\}$ and $E(s, S \setminus C) = \{e_2, e_3\}$, where $\delta(e_1) = [0.6, 0.8]$, $\delta(e_2) = [0, 0.2]$ and $\delta(e_3) = [0, 0.2]$. Then condition (1) holds but condition (2) does not: indeed, at least total probability 0.2 must be assigned to the edges ($e_2$ and $e_3$) that leave $C$. Now consider an IMC with state $s \in C$ such that $E(s, C) = \{e_1, e_2\}$ and $E(s, S \setminus C) = \{e_3\}$, where $\delta(e_1) = [0, 0.5]$, $\delta(e_2) = [0, 0.5]$ and $\delta(e_3) = [0.1, 0.5]$. Then condition (2) holds (because the sum of the right endpoints of the intervals associated with $e_1$ and $e_2$ is equal to 1), but condition (1) does not (because the interval associated with $e_3$ specifies that probability at least 0.1 must be assigned to leaving $C$). Note also that if $E(s, C) \subseteq E^{[\cdot, \cdot)} \cup E^{(\cdot, \cdot)}$ (all edges in $E(s, C)$ have right-open intervals) and $\sum_{e \in E(s,C)} \mathsf{right}(\delta(e)) = 1$, there must exist at least one edge in $E(s, S \setminus C)$ by well formedness. □

Let $\mathcal{I}$ be the set of ILECs of $\mathcal{O}$. We say that an ILEC $C \in \mathcal{I}$ is *maximal* if there does not exist any $C' \in \mathcal{I}$ such that $C \subset C'$. For a path $\rho \in \mathit{Paths}_\infty^{[\mathcal{O}]_I}(s)$, let $\mathit{infst}(\rho) \subseteq S$ be the states that appear infinitely often along $\rho$, i.e., for $\rho = s_0 \mu_0 s_1 \mu_1 \cdots$, we have $\mathit{infst}(\rho) = \{s \in S | \forall i \in \mathbb{N}. \exists j > i. s_j = s\}$. We present a result for ILECs that is analogous to the fundamental theorem of end components of *de Alfaro (1997)*: the result specifies that, with probability 1, a scheduler of the IMDP semantics of $\mathcal{O}$ must confine itself to an ILEC.

**Lemma 9**  *For $s \in S$ and $\sigma \in \Sigma^{[\mathcal{O}]_I}$, we have $\mathrm{Pr}_s^\sigma(\{\rho | \mathit{infst}(\rho) \in \mathcal{I}\}) = 1$.* □

**Proof**  The proof is structured in the same manner as that for classical end components in *de Alfaro (1997)*. Consider $C \subseteq S$ such that $C \notin \mathcal{I}$. Our aim is to show that $\mathrm{Pr}_s^\sigma(\{\rho | \mathit{infst}(\rho) = C\}) = 0$. Given that $\mathcal{I}$ is a finite set, the required result then follows.

First suppose that the condition (1) in the definition of ILECs does not hold, i.e., there exists $(s', s'') \in E$ such that $s' \in C$, $s'' \notin C$ and $(s', s'') \in E^{\langle +, \cdot \rangle}$. Observe that $\mathsf{left}(\delta(s', s'')) > 0$. The probability of remaining in $C$ when visiting $s'$ is at most $1 - \mathsf{left}(\delta(s', s''))$, where $1 - \mathsf{left}(\delta(s', s''))$ is strictly less than 1. Given that $s' \in \mathit{infst}(\rho)$ for every $\rho$ such that $\mathit{infst}(\rho) = C$, we have that $\mathrm{Pr}_s^\sigma(\{\rho | \mathit{infst}(\rho) = C\}) \leq \lim_{k \to \infty} (1 - \mathsf{left}(\delta(s', s'')))^k = 0$.

Suppose that condition (2) in the definition of ILECs does not hold for some $s' \in C$, i.e., $\sum_{e \in E(s',C)} \mathsf{right}(\delta(e)) < 1$. Therefore, for all $\mu \in \Delta(s')$, we must have $\sum_{s'' \in C} \mu(s'') \leq$

$\sum_{e \in E(s',C)} \mathsf{right}(\delta(e)) < 1$. Hence, the probability of remaining in $C$ when visiting $s'$ is strictly less than 1. Then, as in the case of the first condition in the definition of ILECs, we conclude that $\mathrm{Pr}_s^\sigma(\{\rho | infst(\rho) = C\}) \leq \lim_{k \to \infty}(\sum_{e \in E(s',C)} \mathsf{right}(\delta(e)))^k = 0$.

Now suppose that the condition (3) in the definition of ILECs does not hold, *i.e.*, there is no path from state $s'$ to state $s''$ in the graph induced by $C$, where all states along the path (including $s'$ and $s''$) belong to $C$. Given that $S$ is finite, from some point onwards, any path $\rho$ features only those states from $infst(\rho)$. Then for any occurrence of $s'$ in the suffix of $\rho$ that features only states from $infst(\rho)$, there cannot be a subsequent occurrence of $s''$ along the path. Hence we must have $infst(\rho) \neq C$, and thus $\mathrm{Pr}_s^\sigma(\{\rho | infst(\rho) = C\}) = 0$. ∎

We now show that there exists a scheduler that, from a state within an ILEC, can confine the IMC to the ILEC with positive probability. This result is the ILEC analogue of a standard result for end components of finite MDPs that specifies that there exists a scheduler that, from a state of an end component, can confine the MDP to the end component with probability 1 (see *de Alfaro (1997)* and *Baier & Katoen (2008)*). In the case of IMCs and ILECs, it is not possible to obtain an analogous result for probability 1; in the example of Fig. 1, the singleton set $\{s_0\}$ is an ILEC, but it is not possible to find a scheduler that remains in $s_0$ with probability 1, because with each transition the IMC moves to $s_1$ with positive probability. However, for our purposes, it is sufficient to have a result stating that, starting from a state within an ILEC, the IMC can be confined to that ILEC with positive probability.

**Lemma 10** *Let $C \in \mathcal{I}$ and $s \in C$. There exists a scheduler $\sigma \in \Sigma^{[\mathcal{O}]_I}$ such that $\mathrm{Pr}_s^\sigma(\{\rho | \rho \notin \mathsf{Reach}(S \setminus C) \wedge infst(\rho) = C\}) > 0$.* □

**Proof** The intuition underlying the proof of Lemma 10 is the definition of a scheduler that assigns progressively decreasing probability to all edges in $E^{\langle 0, \cdot \rangle}$ that leave ILEC $C$, in such a way as to guarantee that the IMC is confined in $C$ with positive probability. This is possible because conditions (1) and (2) of the definition of ILECs specify that there is no fixed lower bound on the probability that must be assigned to edges that leave $C$. Furthermore, the scheduler is defined so that the remaining probability at each step that is assigned between all edges that stay in $C$ is always no lower than some fixed lower bound; this characteristic of the scheduler, combined with the fact that we remain in $C$ with positive probability and the fact that $C$ is strongly connected, means that we visit all states of $C$ with positive probability under the defined scheduler. The scheduler will depend only on the current state and the number of transitions done so far. In the following, we assume that there exists at least one state $s' \in C$ such that $E^{\langle 0, \cdot \rangle}(s', S \setminus C)$ is non-empty, and consider the case in which this assumption does not hold at the end of the proof. We define the scheduler $\sigma \in \Sigma^{[\mathcal{O}]_I}$ according to the following approach: first we define a lower bound on the probabilities that are assigned by the scheduler $\sigma$ to edges in $E^{\langle 0, \cdot \rangle}$ that have source and target states in $C$; second we define a sequence of probabilities that depend on the number of transitions done so far that is used by $\sigma$ in the assignment of probability to edges in $E^{\langle 0, \cdot \rangle}$ that have source in $C$ and target states not in $C$. Both of the aforementioned probabilities are sufficiently small as to allow the

definition of the assignment chosen by $\sigma$ for each state in $C$ and each natural number corresponding to the number of transitions done so far.

First, we define a constant probability that we will use subsequently to obtain a lower bound on the probability that is assigned by the scheduler $\sigma$ to each edge in $E^{\langle 0, \cdot \rangle}$ that has source and target states in $C$. Let $\tilde{C}$ be the set of states $s' \in C$ such that $E^{\langle 0, \cdot \rangle}(s', C)$ is not empty. In order to define a probability that is sufficiently small in order to allow the definition of an assignment, for each state $s' \in \tilde{C}$ of the ILEC, there are two factors to take into account. On the one hand, the supremum total probability that can be assigned to edges in $E^{\langle 0, \cdot \rangle}(s', C)$ is $1 - \sum_{e \in E(s')} \mathsf{left}(\delta(e))$, *i.e.*, at least probability $\sum_{e \in E(s')} \mathsf{left}(\delta(e))$ must be assigned to edges in $E(s')$ that do *not* have 0 as their left endpoint. Taking into consideration also the fact that we are aiming to identify a *constant* probability for all edges in $E^{\langle 0, \cdot \rangle}(s', C)$, this means that this probability should not exceed $\frac{1 - \sum_{e \in E(s')} \mathsf{left}(\delta(e))}{|E^{\langle 0, \cdot \rangle}(s', C)|}$. On the other hand, and more straightforwardly, the probability assigned to any edge cannot exceed the edge's right endpoint. This intuition underlies the following definition of $\kappa_{s'}$:

$$\kappa_{s'} = \min \left\{ \min_{e \in E^{\langle 0, \cdot \rangle}(s', C)} \mathsf{right}(\delta(e)), \frac{1 - \sum_{e \in E(s')} \mathsf{left}(\delta(e))}{|E^{\langle 0, \cdot \rangle}(s', C)|} \right\}.$$

Let $\overline{\kappa} = \frac{1}{2} \min_{s' \in \tilde{C}} \kappa_{s'}$. We will subsequently use $\overline{\kappa}$ to define a lower bound on the probability assigned to each edge in $E^{\langle 0, \cdot \rangle}(s', C)$ by the distribution chosen by $\sigma$ from paths ending in state $s' \in \tilde{C}$.

Next, we consider the definition of a constant that we will subsequently use to define a sequence of probabilities that will be used to define assignments chosen by $\sigma$ from states in $C$ that have outgoing edges with left endpoint 0 and target state *not* belonging to $C$. Let $\hat{C}$ be the set of states $s' \in C$ such that $E^{\langle 0, \cdot \rangle}(s', S \setminus C)$ is not empty. For $s' \in \hat{C}$, and using intuition similar to that presented in the previous paragraph, we let:

$$\lambda_{s'} = \min \left\{ \min_{e \in E^{\langle 0, \cdot \rangle}(s', S \setminus C)} \mathsf{right}(\delta(e)), \frac{1 - \overline{\kappa} - \sum_{e \in E(s')} \mathsf{left}(\delta(e))}{|E^{\langle 0, \cdot \rangle}(s', S \setminus C)|} \right\}.$$

Intuitively, $\lambda_{s'}$ is the supremum probability that can be assigned to edges with left endpoint 0 from state $s'$ to states *not* in $C$, assuming that probability $\overline{\kappa}$ is assigned to edges with left endpoint 0 that remain in $C$ (note that this assumption is conservative, because in some states there may be no outgoing edges with left endpoint 0 that remain in $C$). Let $\overline{\lambda} = \frac{1}{2} \min_{s' \in \hat{C}} \lambda_{s'}$, and let $\mathcal{I}$ equal the minimum index $i \geq 2$ such that $\frac{1}{2^i} < \overline{\lambda}$. Given $s' \in C$ and $i \in \mathbb{N}$, let $\mathbf{a}_{s'}^i$ be an assignment for $s'$ defined in the following way:

- If $E^{\langle 0, \cdot \rangle}(s', S \setminus C) \neq \emptyset$, then $\mathbf{a}_{s'}^i(s'') = \frac{1}{2^{\mathcal{I}+i}}$ for each $s'' \in S \setminus C$ such that $(s', s'') \in E^{\langle 0, \cdot \rangle}(s', S \setminus C)$.
- If $E^{\langle 0, \cdot \rangle}(s', C) \neq \emptyset$, then $\mathbf{a}_{s'}^i(s'') \geq \overline{\kappa}$ for each $s'' \in C$ such that $(s', s'') \in E^{\langle 0, \cdot \rangle}(s', C)$.

The definitions of $\overline{\kappa}$ and $\overline{\lambda}$ allow us to complete the definition of $\mathbf{a}_{s'}^i$ for states not considered in these two points above, so that $\mathbf{a}_{s'}^i$ is an assignment for $s'$. The intuition here is as follows. Condition (2) of the definition of ILECs specifies that obtaining an assignment that obeys the above constraints and which sums to 1 is possible; the definitions of $\overline{\kappa}$

and $\bar{\lambda}$ guarantee that enough probability mass is available to edges that do *not* have a left endpoint of 0 to equal or exceed their left endpoints, and that the probability that is assigned to edges that *do* have a left endpoint of 0 does not exceed their right endpoint. We now formalise the definition of $\sigma$. Let $r \in Paths_*^{[\mathcal{O}]_I}(s)$, where $r = s_0 \mu_0 s_1 \mu_1 \cdots \mu_{n-1} s_n$ and $s_i \in C$ for all $i \leq n$. We let $\sigma(r) = \{\mathbf{a}_{s_n}^n \mapsto 1\}$.

It remains to show that $\Pr_s^\sigma(\{\rho | \rho \notin \mathsf{Reach}(S \setminus C) \wedge infst(\rho) = C\}) > 0$. Given state set $X \subseteq S$ and $k \in \mathbb{N}$, we let $Reach^{\leq k}(X)$ be the set of paths that reach $X$ within $k$ transitions; formally, $Reach^{\leq k}(X) = \{\rho \in Paths_\infty^{[\mathcal{O}]_I}(s) | \exists i \leq k . \mathsf{state}[\rho](i) \in X\}$. By definition, we have $\mathsf{Reach}(X) = \bigcup_{k \in \mathbb{N}} Reach^{\leq k}(X)$, and $\Pr_s^\sigma(\mathsf{Reach}(X)) = \lim_{k \to \infty} \Pr_s^\sigma(Reach^{\leq k}(X))$. From the fact that $s \in C$, we have $\Pr_s^\sigma(Reach^{\leq 0}(S \setminus C)) = 0$. From the definition of $\sigma$ above, for $k \geq 1$, we have $\Pr_s^\sigma(Reach^{\leq k}(S \setminus C)) \leq \frac{1}{4} + \sum_{i=1}^k \frac{1}{2^{i+2}} \prod_{j=0}^{i-1}(1 - \frac{1}{2^{i+1}}) \leq \frac{1}{4} + \sum_{i=1}^k \frac{1}{2^{i+2}} = \sum_{i=0}^k \frac{1}{2^{i+2}}$. Observe that $\sum_{i=0}^k \frac{1}{2^{i+2}} \leq \frac{1}{2}$ for all $k \in \mathbb{N}$. Hence $\Pr_s^\sigma(\mathsf{Reach}(S \setminus C)) = \lim_{k \to \infty} \Pr_s^\sigma(Reach^{\leq k}(S \setminus C)) \leq \lim_{k \to \infty} \sum_{i=0}^k \frac{1}{2^{i+2}} \leq \frac{1}{2}$. Therefore we have shown that $\Pr_s^\sigma(\{\rho | \rho \notin \mathsf{Reach}(S \setminus C)\}) \geq \frac{1}{2} > 0$.

Because the assignments used by $\sigma$ dedicate a probability value to all edges with source and target states in $C$ that is no lower than some fixed lower bound, and because the graph $(C, E \cap (C \times C))$ is strongly connected, we have that $\Pr_s^\sigma(\{\rho | \rho \notin \mathsf{Reach}(S \setminus C) \wedge infst(\rho) = C\}) = \Pr_s^\sigma(\{\rho | \rho \notin \mathsf{Reach}(S \setminus C)\})$. Hence we have shown that $\Pr_s^\sigma(\{\rho | \rho \notin \mathsf{Reach}(S \setminus C) \wedge infst(\rho) = C\}) > 0$.

Finally, we consider the case in which all states $s' \in C$ are such that $E^{\langle 0, \cdot \rangle}(s', S \setminus C) = \emptyset$. Consider state $s' \in C$. Given that $E^{\langle 0, \cdot \rangle}(s', S \setminus C)$ is empty, then, because condition (1) in the definition of ILECs specifies that $E^{\langle +, \cdot \rangle}(s, S \setminus C)$ is empty, we have that $E(s', S \setminus C) = E^{\langle 0, \cdot \rangle}(s', S \setminus C) \cup E^{\langle +, \cdot \rangle}(s, S \setminus C)$ is empty. Hence, if $E^{\langle 0, \cdot \rangle}(s', S \setminus C)$ is empty for all $s' \in C$, there is no edge $(s', s'') \in E$ such that $s' \in C$ and $s'' \notin C$, and the graph $(C, E \cap (C \times C))$ is a bottom strongly connected component. From this fact, for any scheduler, the probability of remaining in the ILEC is 1. It then remains to define a scheduler $\sigma$ that selects only assignments that assign fixed, positive probability to edges in $E^{\langle 0, \cdot \rangle}$ that have source states in $C$ (note that these edges remain in $C$) in a similar manner as in the previous case of this proof. We can then conclude that $\Pr_s^\sigma(\{\rho | \rho \notin \mathsf{Reach}(S \setminus C) \wedge infst(\rho) = C\}) = 1 > 0$. ∎

**Remark 3** Note that we can confine the IMC to an ILEC with probability 1 if the only edges leaving the ILEC belong to $E^{[0, \cdot \rangle}$; however, we do not require that result, and above we settle for a simplified scheduler construction that does not distinguish between edges in $E^{[0, \cdot \rangle}$ and $E^{(0, \cdot \rangle}$, and hence allows some progressively decreasing probability of exiting from the ILEC even if the outgoing edges of the ILEC are only from $E^{[0, \cdot \rangle}$.. □

Let $U_{\neg T} = \bigcup \{C \in \mathcal{I} | C \cap T = \emptyset\}$ be the union of the ILECs that do not contain states in $T$. Using Lemma 9 and Lemma 10, we can show that the existence of a scheduler of $[\mathcal{O}]_I$ that reaches $T$ with probability strictly less than 1 is equivalent to the existence of a path in the graph of $\mathcal{O}$ that reaches $U_{\neg T}$.

**Proposition 3** Let $s \in S$. There exists $\sigma \in \Sigma^{[\mathcal{O}]_I}$ such that $\Pr_s^\sigma(\mathsf{Reach}(T)) < 1$ if and only if there exists a finite path $r \in Paths_*^{\mathcal{O}}(s)$ such that $last(r) \in U_{\neg T}$. □

**Proof** ($\Rightarrow$) Let $\sigma \in \Sigma^{[\mathcal{O}]_I}$ be such that $\Pr_s^\sigma(\mathsf{Reach}(T)) < 1$. Then, by duality, we have $\Pr_s^\sigma(\{\rho | \rho \notin \mathsf{Reach}(T)\}) > 0$. From this fact, and from Lemma 9, we obtain $\Pr_s^\sigma(\{\rho | \rho \notin \mathsf{Reach}(T) \wedge \mathit{infst}(\rho) \in \mathcal{I}\}) > 0$. For any path $\rho \in \mathit{Paths}_\infty^\sigma(s)$, we have that $\rho \notin \mathsf{Reach}(T)$ and $\mathit{infst}(\rho) \in \mathcal{I}$ implies that $\rho \in \mathsf{Reach}(U_{\neg T})$ from the definition of $U_{\neg T}$. Hence $\Pr_s^\sigma(\{\rho | \rho \in \mathsf{Reach}(U_{\neg T})\}) > 0$. From Lemma 7, there exists a finite path $r \in \mathit{Paths}_*^{\mathcal{O}}(s)$ such that $\mathit{last}(r) \in U_{\neg T}$, completing this direction of the proof.

($\Leftarrow$) The existence of a path $r \in \mathit{Paths}_*^{\mathcal{O}}(s)$ such that $\mathit{last}(r) \in U_{\neg T}$ implies that there exists $\sigma \in \Sigma^{[\mathcal{O}]_I}$ such that $\Pr_s^\sigma(\{\rho | \rho \in \mathsf{Reach}(U_{\neg T})\}) > 0$ by Lemma 7. We define $\sigma' \in \Sigma^{[\mathcal{O}]_I}$ such that $\Pr_s^{\sigma'}(\mathsf{Reach}(T)) < 1$ in the following way: the scheduler $\sigma'$ behaves as $\sigma$ until a state in $U_{\neg T}$ is reached; once such a state has been reached, the scheduler then behaves as a scheduler defined in Lemma 10, which ensures that the states of an ILEC that does not contain any state from $T$ are henceforth visited infinitely often with positive probability. The construction of the scheduler is standard, and we describe it here for completeness. Let $s' \in U_{\neg T}$, and let $C_{s'} \in \mathcal{I}$ be the maximal ILEC such that $s' \in C_{s'}$ and $C_{s'} \cap T = \emptyset$ (which exists by definition of $U_{\neg T}$). Now let $\sigma_{s'} \in \Sigma^{[\mathcal{O}]_I}$ be such that $\Pr_{s'}^{\sigma_{s'}}(\{\rho | \rho \notin \mathsf{Reach}(S \setminus C_{s'}) \wedge \mathit{infst}(\rho) = C_{s'}\}) > 0$, which exists by Lemma 10. For finite paths $r = s_0 \mu_0 s_1 \mu_1 \cdots \mu_{n-1} s_n \in \mathit{Paths}_\infty^{[\mathcal{O}]_I}(s)$ such that $s_i \notin U_{\neg T}$ for all $i \leq n$, we let $\sigma'(r) = \sigma(r)$. For a finite path $r = s_0 \mu_0 s_1 \mu_1 \cdots \mu_{n-1} s_n \in \mathit{Paths}_\infty^{[\mathcal{O}]_I}(s)$ such that there exists $i \leq n$ for which $s_j \notin U_{\neg T}$ for all $j < i$ and $s_j \in U_{\neg T}$ for all $j \geq i$, we let $\sigma'(r) = \sigma_{s_i}(s_i \mu_i \cdots \mu_{n-1} s_n)$. For other finite paths, the definition of $\sigma'$ can be arbitrary. Given that $\Pr_s^\sigma(\{\rho | \rho \in \mathsf{Reach}(U_{\neg T})\}) > 0$, we have $\Pr_s^{\sigma'}(\{\rho | \rho \in \mathsf{Reach}(U_{\neg T})\}) > 0$, and we know that there exists a finite path $r \in \mathit{Paths}_*^{\sigma'}(s)$ (with positive probability) such that $\mathit{last}(r) \in U_{\neg T}$. Given the definition of the behaviour of $\sigma'$ after finite path $r$, we have $\Pr_s^{\sigma'}(\{\rho | r$ is a prefix of $\rho \wedge \rho \notin \mathsf{Reach}(S \setminus C_{\mathit{last}(r)}) \wedge \mathit{infst}(\rho) = C_{\mathit{last}(r)}\}) > 0$. This then implies that $\Pr_s^{\sigma'}(\{\rho | \rho \notin \mathsf{Reach}(S \setminus C_{\mathit{last}(r)}) \wedge \mathit{infst}(\rho) = C_{\mathit{last}(r)}\}) > 0$. Because we have assumed that all states in $T$ are absorbing, the set $T$ is not reached along $r$ (otherwise $U_{\neg T}$ could not contain the final state of $r$); then, given that $C_{\mathit{last}(r)} \cap T = \emptyset$, we have that $\Pr_s^{\sigma'}(\{\rho | \rho \notin \mathsf{Reach}(T)\}) > 0$, and by duality $\Pr_s^{\sigma'}(\mathsf{Reach}(T)) < 1$. Hence this direction of the proof is completed. ∎

Hence we identify the set $S_\forall^{1,I}$ by computing the complement of $S_\forall^{1,I}$, *i.e.*, the set $S \setminus S_\forall^{1,I} = \{s \in S | \forall \sigma \in \Sigma^{[\mathcal{O}]_I} . \Pr_s^\sigma(\mathsf{Reach}(T)) < 1\}$. Using Proposition 3, this set can be computed by considering reachability on the graph of $\mathcal{O}$ of the set $U_{\neg T}$. The set $U_{\neg T}$ can be computed in polynomial time in the size of $\mathcal{O}$ in a manner similar to the computation of maximal end components of MDPs (see *de Alfaro (1997)* and *Baier & Katoen (2008)*). First we compute all strongly connected components $(C_1, E \cap (C_1 \times C_1)), \ldots, (C_m, E \cap (C_m \times C_m))$ of the graph $(S \setminus T, E \cap ((S \setminus T) \times (S \setminus T)))$ of $\mathcal{O}$. Then, for each $1 \leq i \leq m$, we remove from $C_i$ all states for which conditions (1) or (2) in the definition of ILECs do *not* hold with respect to $C_i$ (these conditions can be checked in polynomial time for each state), to obtain the state set $C_i'$. Next, we compute the strongly connected components of the graph $(C_i', E \cap (C_i' \times C_i'))$, and for each of these, repeat the procedure described above. We terminate the algorithm when it is not possible to remove a state (via a failure to satisfy a least one of the conditions (1) and (2) in the definition of

ILECs) from any generated strongly connected component. The generated state sets of the strongly connected components obtained will be the maximal ILECs that do not contain any state in $T$, and their union is $U_{\neg T}$. Hence the overall algorithm for computing $S_{\forall}^{1,I}$ is in polynomial time in the size of $\mathcal{O}$.

# CONCLUSION

We have presented algorithms for qualitative reachability properties for open IMCs. In the context of qualitative properties of system models with *fixed* probabilities on their transitions, probability can be regarded as imposing a fairness constraint, *i.e.*, paths for which a state is visited infinitely often and one of its successors is visited only finitely often have probability 0. In open IMCs, the possibility to make the probability of a transition converge to 0 in the IMDP semantics captures a different phenomenon, which is key for problems concerning the minimum reachability probability over all schedulers being compared to 1. For the three other classes of qualitative reachability problems, we have shown that the UMC and IMDP semantics coincide. We note that the algorithms presented in this article require some numerical computation (a sum and a comparison of the result with 1 in the CPre, APre and ILEC computations), but these operations are simpler than the polynomial-time solutions for quantitative properties of (closed) IMCs in *Chen, Han & Kwiatkowska (2013)* and *Puggelli et al. (2013)*. Similarly, the CPre and APre operators are simpler than the polynomial-time step of value iteration used in the context of quantitative verification of *Haddad & Monmege (2018)*. For the IMDP semantics, our methods give directly a P-complete algorithm for the qualitative fragment of the temporal logic PCTL (*Hansson & Jonsson, 1994*). Future work could consider quantitative properties and $\omega$-regular properties, and applying the results to develop qualitative reachability methods for interval Markov decision processes or for higher-level formalisms such as clock-dependent probabilistic timed automata (*Sproston, 2021a*; *Sproston, 2021b*).

## Funding

The authors received no funding for this work.

## Competing Interests

The authors declare that there are no competing interests.

## Author Contributions

- Jeremy Sproston conceived and designed the experiments, performed the experiments, analyzed the data, performed the computation work, prepared figures and/or tables, authored or reviewed drafts of the article, and approved the final draft.

## Data Deposition

This article does not make reference to data or code.

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
