# Peer review of "Qualitative reachability for open interval Markov chains"

_PeerJ Computer Science, doi:10.7717/peerj-cs.1489_

## Round 0.1 · original submission · Minor Revisions

Your paper has been thoroughly reviewed by four reviewers. Based on their judgement, your submission has to subject to minor revisions. All reviewers clearly appreciate the results for qualitative reachability for *open* interval Markov chains and find the results with to be published. You are requested to take the suggestions by the reviewers into account into your revised paper. In particular, you are requested to highlight the contribution of the paper compared to the RV conference version (R#2), clarify the purpose of the results of Section 2.2 and 2.3 (R#1), highlight the important issues in the proofs of Lemma 2, 4 and 6 (R#4) and to check the relevance of the indicated paper by Delahaye et al. (R#4).

I sincerely hope that the comments of the reviewers are helpful and constructive for a revision, and I am looking forward to receive the revised version. The revised version will be checked by the associate editor, and will not be sent to the reviewers again.

Reviewer 1 ·

Basic reporting

The paper is well-written. Language and definitions are always clear. The aim of the paper is motivated and all results are self-contained.

The introduction sometimes is a bit too technical, e.g. the description of the MDP abstraction starting at line 71 on page 2 (The set of states of the finite MDP that we construct ...).

My suggestion would be to clarify the purpose of the results of Section 2.2 and 2.3. For starters, I'd propose to move them into a new main section (Section 3, then) to stress that these are key ingredients of your approach (and not part of the preliminaries). My intuition is that these results mainly deal with the potential openness (or half-openness) of the intervals. Subsequent sections can then often use terms like "edges" or "valid edge sets" instead which in some sense abstract away from the different types of intervals.

Proposition 1 appears quite fundamental. I wonder if this generalizes any known result for (closed?) IMCs or if this is a new result. In any case, this should be made explicit in the text.

One of your results is that qualitative reachability under the two different semantics mostly coincides, except for almost sure reachability under all probability assignments. I think it is worth mentioning this more explicitly, e.g. as a corollary and also in the introduction of the paper.

Experimental design

The paper clearly states the theoretical problems it concerns itself with and describes algorithms to solve these problems.

Validity of the findings

All lemmas, propositions, and theorems are accompanied by comprehensive correctness proofs. As far as I have checked, these proofs are sound.
The drawn conclusions address the original research question.

Additional comments

- Page 3/line 97: to me, it wasn't clear why the series converges to a value below 1. Some additional argument for this would be helpful.
- Page 5/line 205: "we wil" -> "we will"
- Page 8/line 350: In the chain of inequalities ("We have term1 <= term2 <= term3 <= 1") I think the third term where you sum over all edges e in E should be dropped.
- Page 12/Proof of Lemma 2: I think in the second case (<==) one must assume that w.l.o.g. the considered path has no cycle. Otherwise, the path potentially visits the same state s twice and it would not be clear which assignment to take for s. The proof of lemma 6 is similar.
- Page 12/line 536: In the first line of the proof of Lemma 3, the probability to reach T should be lambda (and not 0)
- Page 14/line 625: "we show that APre can *be* characterised by..."
- Page 19/line 850: "is no greater than" -> "is at most"

Cite this review as

Reviewer 2 ·

Basic reporting

In this paper, the author considers the problem of checking whether a given set of target states T can be reached with probability equal to or different than 0 or 1 in an open interval Markov chain (IMC), i.e., a Markov chain whose transition probabilities are constrained by intervals instead of being numbers. Differently from other works in literature, the intervals can have open boundaries, e.g. (0,0.7).
The author considers the four reachability problems (=0, >0, <1, =1) and two semantics for the IMC (UMC and IMDP) and provides polynomial algorithms for deciding the problem at hand. In the UMC (uncertain Markov chain) semantics, the probability values of the transitions are chosen in the corresponding interval in advance (so each choice corresponds to a classical discrete time Markov chain -- DTMC) and the reachability problem holds if T is reached with the desired probability in each induced DTMC; in the IMDP (interval Markov decision process) semantics, instead, in each state of the IMC all possible probability distributions are kept available, so in successive visits to the same state the outgoing transitions can occur with different probability.

I think the paper is quite interesting and worth of publication; however, my main concern is about the actual contribution with respect to its conference version, published in RP'18. Except for the proofs of the results, the current submission does not add any theoretical contribution to the considered topic. Moreover, the proofs seem rather ordinary without contributing new insights in the considered problem. The only exception is the proof of the algorithm for the IMDP version of the reachability 1 problem (cf. Section 4.3), where a specific scheduler is constructed so that it can remain forever in a set of states with positive probability even if the outgoing transitions must have non-zero probability. However, I am not sure whether this construction suffices to justify a journal version of a conference paper already presenting all the theoretical results.

Experimental design

no comment

Validity of the findings

The results seem reasonable and correct.

Additional comments

Minor remarks:
everywhere: it is really hard to parse correctly sentences when multiple citations occur together, so make sure to clearly mark groups of citations, like by using \pcite{...} or (\cite{...})
62: "of qualitative" -> "of the qualitative"
92: "problem" -> "problems"
100: avoid the space before the footnote's marker
101: "problem" -> "problems"
336: the set E^{⟨·,·]} is not defined (due to ⟨·,·] not being introduced)
408: "forms" -> "form"
409: maybe "or" -> "or to"
527: "asignment" -> "assignment"
625: "can" -> "can be"
771: "which is" -> "which are" (to link with "end components")
837: "a" -> "at"
875: "ı̀ntuition" -> "intuition"
905/913: I would use larger curly brackets, either by using an appropriate version of \big/\Big/... or by using \left/\right
947: "the probability of remaining in the ILEC has probability 1" -> "the probability of remaining in the ILEC is 1" or "remaining in the ILEC has probability 1"
956: "union of states of ILECs" -> "union of ILECs" or "union of set of states of ILECs"
971: something is missing in "the states ILEC that does not contain any state", maybe "states of an ILEC"
983: "could not be" -> "could not contain"
996: "faliure" -> "failure"
998: "be be" -> "be"
1014: "Pctl" -> "PCTL"

Cite this review as

Reviewer 3 ·

Basic reporting

The paper under consideration is involved with interval Markov chains (IMCs) with potentially open intervals. IMCs are a variant of Markov models in which probabilities are not specified precisely as single numbers. Rather, transitions are equipped with intervals, which are assumed to contain the true transition probability. Depending on the interpretation used, it is either assumed that once a choice about the concrete probability is made, or that a concrete probability is chosen each time the transition is taken.

Most works about IMC consider intervals which are closed. The paper under consideration is involved with IMCs the transition intervals of which are potentially open. The paper is then involved with solving the question of whether it is possible to reach a set of target states with a probability arbitrarily close to 0 or 1.

I propose to accept the paper. As far as I am aware, the problem has not been considered before and is interesting to the community. All claims made are accompagnied by according proofs. The results are correct to the best of my confidence. The paper is readable.

Experimental design

no comment

Validity of the findings

no comment

Additional comments

no comment

Cite this review as

·

Basic reporting

- the definition of a valid set of states should be properly highlighted in a Definition environment that also contains the notions of realizable and large sets of states.

- the definition of an IMC-level end component should be properly highlighted in a Definition environment.

- I suggest the author consider the paper entitled "Reachability in Parametric Interval Markov Chains using Constraints" by Bart, Delahaye, Fournier, Lime, Monfroy, and Truchet. In particular, the paper establishes the equivalence of the UMC and IMDP semantics w.r.t. reachability properties for (closed) interval Markov chains.

Experimental design

no comment

Validity of the findings

- Proof of Lemma 2 (<=) should assume w.l.o.g. that the path s0...sn is simple (i.e., does not contains cycles). This ensures that the selection of the assignments made for each s_i does not need to take into consideration that the same assignment is taken for each repeated state. Additionally, the proof should explicitly say that states that are not present in the path can be associated with any assignment.

- proof of Lemma 4 is based on the observation that for any B,B' in E(s,X) such that B \subseteq B', if B is valid, then B' is. Since this point is crucial for the proof, it would be helpful to have this fact proven in detail (in particular the case when B is large because case (b) applies).

- proof of Lemma 6 (<=) the path s0...sn should be also assumed to be simple, for the same reason mentioned above.

Additional comments

List of Typos & minor comments:

- line 336: it is used E^{<.,.]} but <.,.] was not listed in the set \star defined in line 330.
- line 527: asignment --> assignment
- line 603: you use the property \overline{CPre}(X) = S \ CPre(S \ X) but in line 570 you mention the other equality of the duality. Both equalities are equivalent, however, I suggest introducing in line 570 the one that will be actually used later on.
- line 625: APre can (be) characterized
- proof of Lemma 7 can be simplified by invoking Lemma 2 and the fact that a DTMC in [O]_U actually describes a (memoryless) scheduler \sigma in \Sigma^{[O]_I}.
- line 875: ìntuition --> intuition

Possible suggestions to improve the overall presentation:
- Proposition 1, Proposition 2, and Lemma 1 have very long proofs. Maybe these proofs can be moved to an appendix and in the main text you could just put proof sketches. This would greatly ease the navigation in the paper when one needs to move back to recall some definitions/results.
- I suggest merging Lemma 2 and 7 as well as Lemma 3 and 8 and explicitly presenting the fact that the UMC and the IMDP semantics are equivalent w.r.t. those kinds of reachability queries. Then the discussion on the complexity can be unified.

Cite this review as

---

## Round 0.2 · accepted · Accept

I am happy to see that the major comments of the reviewers have been taken into account in this version of the paper, and we are all satisfied with these adjustments. The paper is ready to be published. Congratulations on the acceptance of your paper.